# Interventional Causal Discovery in a Mixture of DAGs

**Burak Varıcı**[*]
Carnegie Mellon University

**Dmitriy A. Katz**
IBM Research

**Dennis Wei**
IBM Research

**Prasanna Sattigeri**
IBM Research

**Ali Tajer**
Rensselaer Polytechnic Institute

## Abstract

Causal interactions among a group of variables are often modeled by a single causal graph. In some domains, however, these interactions are best described by multiple co-existing causal graphs, e.g., in dynamical systems or genomics. This paper addresses the hitherto unknown role of interventions in learning causal interactions among variables governed by a mixture of causal systems, each modeled by one directed acyclic graph (DAG). Causal discovery from mixtures is fundamentally more challenging than single-DAG causal discovery. Two major difficulties stem from (i) an inherent uncertainty about the skeletons of the component DAGs that constitute the mixture and (ii) possibly cyclic relationships across these component DAGs. This paper addresses these challenges and aims to identify edges that exist in at least one component DAG of the mixture, referred to as the *true* edges. First, it establishes matching necessary and sufficient conditions on the size of interventions required to identify the true edges. Next, guided by the necessity results, an adaptive algorithm is designed that learns all true edges using $\mathcal{O}(n^2)$ interventions, where $n$ is the number of nodes. Remarkably, the size of the interventions is optimal if the underlying mixture model does not contain cycles across its components. More generally, the gap between the intervention size used by the algorithm and the optimal size is quantified. It is shown to be bounded by the *cyclic complexity number* of the mixture model, defined as the size of the minimal intervention that can break the cycles in the mixture, which is upper bounded by the number of cycles among the ancestors of a node.

## 1 Introduction

The causal interactions in a system of causally related variables are often abstracted by a directed acyclic graph (DAG). This is the common practice in various disciplines, including biology [1], social sciences [2], and economics [3]. In a wide range of applications, however, the complexities of the observed data cannot be reduced to conform to a single DAG, and they are best described by a mixture of multiple co-existing DAGs over the same set of variables. For instance, gene expression of certain cancer types comprises multiple subtypes with different causal relationships [4]. In another example, mixture models are often more accurate than unimodal distributions in representing dynamical systems [5], including time-series trajectories in psychology [6] and data from complex robotics environments [7].

Despite the widespread applications, causal discovery for a mixture of DAGs remains an under-investigated domain. Furthermore, the existing studies on the subject are also limited to using only observational data [8–11]. Observational data alone is highly insufficient in uncovering causal relationships. It is well-established that even for learning a single DAG, observational data can learn a DAG only up to its Markov equivalence class (MEC) [12]. Hence, *interventions*, which refer to

---

[*]Work was done when BV was a Ph.D. student at Rensselaer Polytechnic Institute.

38th Conference on Neural Information Processing Systems (NeurIPS 2024).

altering the causal mechanisms of a set of target nodes, have a potentially significant role in improving identifiability guarantees in mixture DAG models. Specifically, interventional data can be used to learn specific cause-effect relationships and refine the equivalence classes.

Using interventions for learning a single DAG is well-investigated for various causal models and interventions [13–17]. In this paper, we investigate using interventions for causal discovery in a mixture of DAGs, a fundamentally more challenging problem. The major difficulties stem from (i) an inherent uncertainty about the skeletons of the DAGs that constitute the mixture and (ii) possibly cyclic relationships across these DAGs. For a single DAG, the skeleton can be learned from observational data via conditional independence (CI) tests and the role of interventions is limited to orienting the edges. On the contrary, in a mixture of DAGs, the skeleton cannot be learned from observational data alone, making interventions essential for both learning the skeleton and orienting the edges. Uncertainty in the skeleton arises because, in addition to *true edges* present in at least one individual DAG, there are *inseparable* random variable pairs that cannot be made conditionally independent via CI tests, even though they are nonadjacent in every DAG of the mixture. These types of inseparable node pairs, referred to as *emergent edges* [11], cannot be distinguished from true edges using observational data alone.

In this paper, we aim to characterize the fundamental limits of interventions needed for learning the true edges in a mixture of DAGs. The two main aspects of these limits are the minimum *size* and *number* of the interventions. To this end, we first investigate the necessary and sufficient size of interventions for identifying a true edge. Subsequently, we design an adaptive algorithm that learns the true edges using interventions guided by the necessary and sufficient intervention sizes. We quantify the optimality gap of the maximum intervention size used by the algorithm as a function of the structure of the cyclic relationships across the mixture model. We note that the component DAGs of the mixture cannot be identified without further assumptions even when using interventions (see examples in Appendix D.1). Hence, our focus is on learning the set of true edges in the mixture as specified above. Our contributions are summarized as follows.

- **Intervention size:** We establish matching necessary and sufficient intervention size to identify each node's *mixture parents* (i.e., the union of its parents across all DAGs). Specifically, we show that this size is one more than the number of mixture parents of the said node.

- **Tree DAGs:** For the special case of a mixture of directed trees, we show that the necessary and sufficient intervention size is one more than the number of DAGs in the mixture.

- **Algorithm:** We design an adaptive algorithm that identifies all directed edges of the individual DAGs in the mixture by using $\mathcal{O}(n^2)$ interventions, where $n$ is the number of variables. Remarkably, the maximum size of the interventions used in our algorithm is optimal if the mixture ancestors of a node (i.e., the union of its ancestors across all DAGs) do not form a cycle.

- **Optimality gap:** We show that the gap between the maximum intervention size used by the proposed algorithm for a given node and the optimal size is bounded by the *cyclic complexity number* of the node, which is defined as the number of nodes needing intervention to break cycles among the ancestors of the node, and is upper bounded by the number of such cycles.

We provide an overview of the closely related literature, the majority of which is focused on the causal discovery of single DAGs.

**Causal discovery of a mixture of DAGs.** The relevant literature on the causal discovery of a mixture of DAGs focuses on developing graphical models to represent CI relationships in the observed mixture distribution [8–11]. Among them, [8] proposes a *fused graph* and shows that the mixture distribution is Markov with respect to it. The study in [9] proposes a similar mixture graph but relies on longitudinal data to orient any edges. The study in [10] constructs a *mixture DAG* that represents the mixture distribution and designed an algorithm for learning a maximal ancestral graph. The algorithm of [10] requires the component DAGs of the mixture to be poset compatible, which rules out any cyclic relationships across the DAGs. The study in [11] introduces the notion of *emergent edges* to investigate the inseparability conditions arising in the mixture of DAGs. The study in [18] proposes a variational inference-based approach for causal discovery from a mixture of time-series data. Despite their differences, all these studies are limited to using observational data.

**Intervention design for causal discovery of a single DAG.** We note that the structure of a single DAG without latent variables can be learned using single-node interventions. Hence, the majority of the literature focuses on minimizing the number of interventions. Worst-case bounds on the number

of interventions with unconstrained size are established in [13], and heuristic adaptive algorithms are proposed in [14]. Intervention design on causal graphs with latent variables is studied in [19–21]. The study in [20] also shows that single-node interventions are not sufficient for exact graph recovery in the presence of latent variables. In another direction, [16] studies interventions under size constraints, establishes a lower bound for the number of interventions, and shows that $\mathcal{O}(\frac{n}{k} \log \log k)$ randomized interventions with size $k$ suffice for identifying the DAG with high probability. In the case of single-node interventions, adaptive and non-adaptive algorithms are proposed in [22], active learning of directed trees is studied in [23], and a universal lower bound for the number of interventions is established in [17]. [24] also studies the universal lower bound problem and [25] provides an exact characterization for the number of interventions required to recover the DAG from the observational essential graph. A linear cost model, where the cost of an intervention is proportional to its size, is proposed in [26]. It is shown that learning the DAG with optimal cost under the linear cost model is NP-hard [27]. The size of the minimal intervention sets is studied for cyclic directed models in [28]. Specifically, it is shown that the required intervention size is at least $\zeta - 1$ where $\zeta$ denotes the size of the largest strongly connected component in the cyclic model. A related problem to intervention design is causal discovery from a combination of observational and interventional data. In this setting, the characterization of the equivalence classes and designing algorithms for learning them is well-explored for a single DAG [29–32].

**Causal discovery from multiple clusters/contexts.** Another approach to causal discovery from a mixture of DAGs is clustering the observed samples and performing structure learning on each cluster separately [33–37]. Learning from multiple contexts is also studied in the interventional causal discovery literature [38–41]. However, these studies assume that domain indexes are known. In a similar problem, [42] aims to learn the domain indexes and perform causal discovery simultaneously.

## 2 Preliminaries and definitions

### 2.1 Observational mixture model

**DAG models.** We consider $K \geq 2$ distinct DAGs $\mathcal{G}_\ell \triangleq (\mathbf{V}, \mathbf{E}_\ell)$ for $\ell \in \{1, \ldots, K\}$ defined over the same set of nodes $\mathbf{V} \triangleq \{1, \ldots, n\}$. $\mathbf{E}_\ell$ denotes the set of *directed* edges in graph $\mathcal{G}_\ell$. Throughout the paper, we refer to these as the mixture *component* DAGs. We use $\mathrm{pa}_\ell(i)$, $\mathrm{ch}_\ell(i)$, $\mathrm{an}_\ell(i)$, and $\mathrm{de}_\ell(i)$ to refer to the parents, children, ancestors, and descendants of node $i$ in DAG $\mathcal{G}_\ell$, respectively. We also use $i \overset{\ell}{\rightsquigarrow} j$ to denote $i \in \mathrm{an}_\mathrm{m}(j)$. For each node $i \in \mathbf{V}$, we define $\mathrm{pa}_\mathrm{m}(i)$ as the union of the nodes that are parents of $i$ in at least one component DAG and refer to $\mathrm{pa}_\mathrm{m}(i)$ as the *mixture parents* of node $i$. Similarly, for each node $i \in \mathbf{V}$ we define $\mathrm{ch}_\mathrm{m}(i)$, $\mathrm{an}_\mathrm{m}(i)$, and $\mathrm{de}_\mathrm{m}(i)$.

**Mixture model.** Each of the component DAGs represents a Bayesian network. We denote the random variable generated by node $i \in \mathbf{V}$ by $X_i$ and define the random vector $X \triangleq (X_1, \ldots, X_n)^\top$. For any subset of nodes $A \subseteq \mathbf{V}$, we use $X_A$ to denote the vector formed by $X_i$ for $i \in A$. We denote the probability density function (pdf) of $X$ under $\mathcal{G}_\ell$ by $p_\ell$, which factorizes according to $\mathcal{G}_\ell$ as

$$p_\ell(x) = \prod_{i \in [n]} p_\ell(x_i \mid x_{\mathrm{pa}_\ell(i)}), \quad \forall \ell \in [K]. \tag{1}$$

For distinct $\ell, \ell' \in [K]$, $p_\ell$ and $p_{\ell'}$ can be distinct even when $\mathbf{E}_\ell = \mathbf{E}_{\ell'}$. The differences between any two DAGs are captured by the nodes with distinct causal mechanisms (i.e., conditional distributions) in the DAGs. To formalize such distinctions, we define the following set, which contains all the nodes with at least two different conditional distributions across component distributions.

$$\Delta \triangleq \left\{ i \in \mathbf{V} : \exists \ell, \ell' \in [K] : p_\ell(X_i \mid X_{\mathrm{pa}_\ell(i)}) \neq p_{\ell'}(X_i \mid X_{\mathrm{pa}_{\ell'}(i)}) \right\}. \tag{2}$$

We adopt the same mixture model as the prior work on causal discovery of mixture of DAGs [8–11, 18]. Specifically, observed data is generated by a mixture of distributions $\{p_\ell : \ell \in [K]\}$. It is unknown to the learner which model is generating the observations $X$. To formalize this, we define $L \in \{1, \ldots, K\}$ as a latent random variable where $L = \ell$ specifies that the true model is $p_\ell$. We denote the probability mass function (pmf) of $L$ by $r$. Hence, we have the following mixture distribution for the observed samples $X$.

$$p_\mathrm{m}(x) \triangleq \sum_{\ell \in [K]} r(\ell) \cdot p_\ell(x). \tag{3}$$

Next, we provide several definitions that are instrumental to formalizing causal discovery objectives.

**Definition 1** (True edge). *We say that $j \to i$ is a* true *edge if $j \in \mathrm{pa_m}(i)$. The set of all true edges is denoted by*

$$\mathbf{E}_\mathrm{t} \triangleq \{(j \to i) : i, j \in \mathbf{V}, \ \exists \, \mathcal{G}_\ell : j \in \mathrm{pa}_\ell(i)\} \,. \tag{4}$$

A common approach to causal discovery is the class of constraint-based approaches, which perform conditional independence (CI) tests on the observed data to infer (partial) knowledge about the DAGs' structure [43–45]. In this paper, we adopt a constraint-based CI testing approach. Following this approach, the following definition formally specifies the set of node pairs that cannot be made conditionally independent in the mixture distribution.

**Definition 2** (Inseparable pair). *The node pair $(i, j)$ is called* inseparable *if $X_i$ and $X_j$ are always statistically dependent in the mixture distribution $p_\mathrm{m}$ under any conditioning set. The set of inseparable node pairs is specified by*

$$\mathbf{E}_\mathrm{i} \triangleq \{(i - j) : i, j \in \mathbf{V}, \ \nexists A \subseteq \mathbf{V} \setminus \{i, j\} : \ X_i \perp\!\!\!\perp X_j \mid X_A \ in \ p_\mathrm{m}\} \,. \tag{5}$$

Note that when $(j \to i)$ is a true edge, the pair $(i, j)$ will be inseparable. A significant difference between independence tests for mixture models and single-DAG models is that not all inseparable pairs have an associated true edge in the former. More specifically, due to the mixing of multiple distributions, a pair of nodes can be nonadjacent in all component DAGs but still be inseparable in mixture distribution $p_\mathrm{m}$. We refer to such inseparable node pairs as *emergent pairs*, formalized next.

**Definition 3** (Emergent pair). *An inseparable pair $(i, j) \in \mathbf{E}_\mathrm{i}$ is called an* emergent pair *if there is no true edge associated with the pair. The set of emergent pairs is denoted by*

$$\mathbf{E}_\mathrm{e} \triangleq \{(i, j) \in \mathbf{E}_\mathrm{i} : \ i \notin \mathrm{pa_m}(j) \ \wedge \ j \notin \mathrm{pa_m}(i)\} \,. \tag{6}$$

The conditions under which emergent edges arise in mixture models are recently investigated in [11], where it is shown that the causal paths that pass through a node in the set $\Delta$ defined in (2) are instrumental for their analysis. These paths are specified next.

**Definition 4** ($\Delta$-through path). *We say that a causal path in $\mathcal{G}_\ell$ between $i$ and $j$ is a $\Delta$-through path if it passes through at least one node in $\Delta$, i.e., there exists $u \in \Delta$ such that $i \overset{\ell}{\rightsquigarrow} u \overset{\ell}{\rightsquigarrow} j$. If $u \in \mathrm{ch}_\ell(i)$, the path is also called a $\Delta$-child-through path.*

## 2.2 Intervention model

In this section, we describe the intervention model we use for causal discovery on a mixture of DAGs. We consider stochastic *hard* interventions on component DAGs of the mixture model. A hard intervention on a set of nodes $\mathcal{I} \subseteq \mathbf{V}$ cuts off the edges incident on nodes $i \in \mathcal{I}$ in all component DAGs $\mathcal{G}_\ell$ for $\ell \in [K]$. We denote the post-intervention component DAGs upon an intervention $\mathcal{I}$ by $\{\mathcal{G}_{\ell,\mathcal{I}} : \ell \in [K]\}$. We note that hard interventions are less restrictive than *do* interventions, which not only remove ancestral dependencies but also remove randomness by assigning constant values to the intervened nodes. Specifically, in $\mathcal{G}_{\ell,\mathcal{I}}$, the causal mechanism of an intervened node $i \in \mathcal{I}$ changes from $p_\ell(x_i \mid x_{\mathrm{pa}_\ell(i)})$ to $q_i(x_i)$. Therefore, upon an intervention $\mathcal{I} \subseteq \mathbf{V}$, the interventional component DAG distributions are given by

$$p_{\ell,\mathcal{I}}(x) \triangleq \prod_{i \in \mathcal{I}} q_i(x_i) \prod_{i \in \mathbf{V} \setminus \mathcal{I}} p_\ell(x_i \mid x_{\mathrm{pa}_\ell(i)}) \,, \qquad \forall \ell \in [K] \,. \tag{7}$$

Subsequently, the interventional mixture distribution $p_{\mathrm{m},\mathcal{I}}(x)$ is given by

$$p_{\mathrm{m},\mathcal{I}}(x) \triangleq \sum_{\ell \in [K]} r(\ell) \cdot p_{\ell,\mathcal{I}}(x) \,. \tag{8}$$

We note that an intervened node $i \in \mathcal{I}$ has the same causal mechanism $q_i(x_i)$ for all interventions $\mathcal{I} \subseteq \mathbf{V}$ that contain $i$. This is because an intervention procedure targets a set of nodes in all mixture components at the same time. Hence, resulting $q_i(X_i)$ is shared for all component models, owing to the same intervention mechanism, e.g., gene knockout experiments [46]. Hence, the set of nodes with distinct causal mechanisms across the components of the interventional mixture model becomes $\Delta_\mathcal{I} \triangleq \Delta \setminus \mathcal{I}$. Next, we specify the $\mathcal{I}$-mixture DAG, which extends the mixture DAG defined for observational data in [10, 11] and will facilitate our analysis.

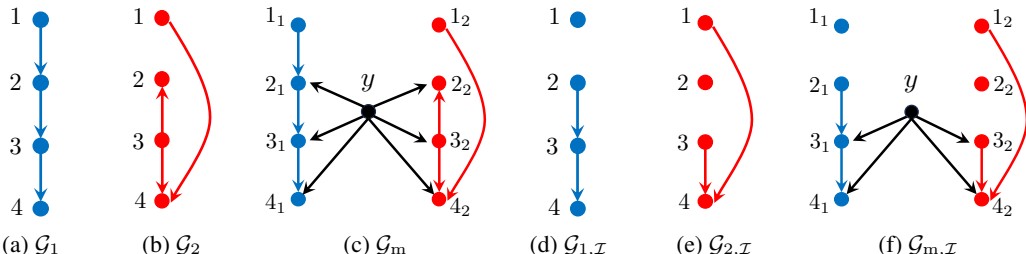

Figure 1: **(a)-(b)**: sample component DAGs; **(c)** the mixture DAG for $\mathcal{I} = \emptyset$, note that $\Delta = \{2, 3, 4\}$ (when the distribution of node 1 remains the same) ; **(d)-(e)**: post-intervention component DAGs for $\mathcal{I} = \{2\}$; **(f)**: corresponding $\mathcal{I}$-mixture DAG. Also note that true edges $\mathbf{E}_{\mathrm{t}} = \{(1 \to 2), (2 \to 3), (3 \to 2), (3 \to 4), (1 \to 4)\}$, inseparable pairs $\mathbf{E}_{\mathrm{i}} = \{(1 - 2), (1 - 3), (1 - 4), (2 - 3), (2 - 4), (3 - 4)\}$, and emergent edges $\mathbf{E}_{\mathrm{e}} = \{(1, 3), (2, 4)\}$.

**Definition 5** ($\mathcal{I}$-mixture DAG). *Given an intervention $\mathcal{I}$ on a mixture of DAGs, $\mathcal{I}$-mixture DAG $\mathcal{G}_{\mathrm{m},\mathcal{I}}$ is a graph with $nK + 1$ nodes constructed by first concatenating the $K$ component DAGs and then adding a single node $y$ to the concatenation. Furthermore, there will be a directed edge from $y$ to every node in $\Delta_{\mathcal{I}}$ in every DAG $\{\mathcal{G}_{\ell,\mathcal{I}} : \ell \in [K]\}$. In the $\mathcal{I}$-mixture DAG $\mathcal{G}_{\mathrm{m},\mathcal{I}}$, we use $i_\ell$ to denote the copy of node $i$ in $\mathcal{G}_{\ell,\mathcal{I}}$. Accordingly, for any $A \subseteq \mathbf{V}$ we define $\bar{A} \triangleq \{i_\ell : i \in A, \ \ell \in [K]\}$.*

Figure 1 illustrates an example of a mixture of $K = 2$ component DAGs, different edge types, an intervention $\mathcal{I}$ on the mixture, and the construction of the $\mathcal{I}$-mixture DAG from post-intervention component DAGs $\mathcal{G}_{1,\mathcal{I}}$ and $\mathcal{G}_{2,\mathcal{I}}$. We define the *observational mixture DAG* as the $\mathcal{I}$-mixture DAG when the intervention set is $\mathcal{I} = \emptyset$ and denote it by $\mathcal{G}_{\mathrm{m}}$. It is known that $p_{\mathrm{m}}$ specified in (3) satisfies the global Markov property with respect to observational mixture DAG [10, Theorem 3.2]. It can be readily verified that this result extends to the interventional setting for $p_{\mathrm{m},\mathcal{I}}$ and $\mathcal{G}_{\mathrm{m},\mathcal{I}}$. We make the following faithfulness assumption to facilitate causal discovery via statistical independence tests.

**Assumption 1** ($\mathcal{I}$-mixture faithfulness). *For any intervention $\mathcal{I} \subseteq \mathbf{V}$, the interventional mixture distribution $p_{\mathrm{m},\mathcal{I}}(x)$ is faithful to $\mathcal{G}_{\mathrm{m},\mathcal{I}}$, that is if $X_A \perp\!\!\!\perp X_B \mid X_C$ in $p_{\mathrm{m},\mathcal{I}}(x)$, then $\bar{A}$ and $\bar{B}$ are d-separated given $\bar{C}$ in $\mathcal{G}_{\mathrm{m},\mathcal{I}}$.*

Finally, we note that the observational counterpart of Assumption 1, i.e., when $\mathcal{I} = \emptyset$, is standard in the literature for analyzing a mixture of DAGs [9–11]. In working with interventions, we naturally extend it to interventional mixture distributions. Also note that Assumption 1 does not compare observational and interventional distributions. Hence, it is not comparable to various faithfulness assumptions in the literature on the interventional causal discovery of a single DAG, e.g., [29, 31].

## 2.3 Causal discovery objectives

We aim to address the following question: *how can we use interventions to perform causal discovery in a mixture of DAGs*, with the objectives specified next.

The counterpart of this question is well-studied for the causal discovery of a single DAG. Since the unoriented skeleton of the single DAG can already be identified by CI tests on observational data, interventions are leveraged to orient the edges. Interventions are generally bounded by a pre-specified budget, measured by the number of interventions. The extent of causal relationships that observational data can uncover in a mixture of DAGs is significantly narrower than those in single DAGs. The striking difference is the existence of emergent pairs specified in (6). Therefore, the objective of intervention design extends to distinguishing *true* cause-effect relationships from the emergent pairs as well as determining the direction of causality. Specifically, we focus on identifying the true edges specified in (4) as the edges exist in at least one component DAG of the mixture. For this purpose, two central objectives of our investigation are:

1. Determining the necessary and sufficient size of the interventions for identifying true edges $\mathbf{E}_{\mathrm{t}}$,

2. Designing efficient algorithms with near-optimal intervention sizes.

# 3    Interventions for causal discovery of a mixture of DAGs

In this section, we investigate the first key question of interventional causal discovery on a mixture of DAGs and investigate the size of the necessary and sufficient interventions for identifying mixture parents of a node. First, we consider a mixture of general DAGs without imposing structural constraints and establish matching necessary and sufficient intervention size for distinguishing a true edge from an emergent pair. Then, we strengthen the results for a mixture of directed trees. The results established in this section are pivotal for understanding the fundamental limits of causal discovery of a mixture of DAGs. These results guide the intervention design in Section 4.

Our analysis uncovers the connections between the mixture distribution under an intervention $\mathcal{I}$ and the structure of post-intervention component DAGs $\{\mathcal{G}_{\ell,\mathcal{I}} : \ell \in [K]\}$. We know that the interventional mixture distribution $p_{\mathrm{m},\mathcal{I}}$ satisfies the Markov property with respect to $\mathcal{I}$-mixture DAG $\mathcal{G}_{\mathrm{m},\mathcal{I}}$ specified in Definition 5. Therefore, in conjunction with the $\mathcal{I}$-mixture faithfulness assumption, the separation statements in $\mathcal{G}_{\mathrm{m},\mathcal{I}}$ can be inferred exactly by testing the conditional independencies in $p_{\mathrm{m},\mathcal{I}}$. To establish the necessary and sufficient intervention sizes, we recall that set $\Delta$ plays an important role in the separability conditions in $\mathcal{I}$-mixture DAG $\mathcal{G}_{\mathrm{m},\mathcal{I}}$ since $\Delta$ allows paths across different component DAGs. The following result serves as an intermediate step in obtaining our main result.

**Lemma 1.** *Consider an inseparable pair $(i, j) \in \mathbf{E}_{\mathrm{i}}$ and an intervention $\mathcal{I} \subseteq \mathbf{V}$. We have the following identifiability guarantees using the interventional mixture distribution $p_{\mathrm{m},\mathcal{I}}(x)$.*

*(i)* **Identifiability:** *It is possible to determine whether $j \in \mathrm{pa}_{\mathrm{m}}(i)$ if $j \in \mathcal{I}$ and there do not exist $\Delta$-through paths from $j$ to $i$ in $\mathcal{G}_{\ell,\mathcal{I}}$ for any $\ell \in [K]$.*

*(ii)* **Non-identifiability:** *It is impossible to determine whether $j \in \mathrm{pa}_{\mathrm{m}}(i)$ if $j \in \Delta_{\mathcal{I}}$ or there exists a $\Delta$-child-through path from $j$ to $i$ in at least one $\mathcal{G}_{\ell,\mathcal{I}}$ where $\ell \in [K]$.*

Lemma 1 provides intuition for characterizing sufficient and necessary conditions for identifying a true edge. The identifiability result implies that it suffices to choose an intervention $\mathcal{I}$ that reduces the viable $\Delta$-through paths in $\mathcal{G}_{\mathrm{m},\mathcal{I}}$ to true edges from $j$ to $i$. Similarly, the non-identifiability result implies the necessity of intervening on $\Delta$-child nodes. Building on these properties, our main result in this section establishes matching necessary and sufficient intervention sizes for identifying true edges.

**Theorem 1** (Intervention sizes). *Consider nodes $i, j \in \mathbf{V}$ in a mixture of DAGs.*

*(i)* **Sufficiency:** *For any mixture of DAGs, there exists an intervention $\mathcal{I}$ with $|\mathcal{I}| \leq |\mathrm{pa}_{\mathrm{m}}(i)| + 1$ that ensures the determination of whether $j \in \mathrm{pa}_{\mathrm{m}}(i)$ using CI tests on $p_{\mathrm{m},\mathcal{I}}$.*

*(ii)* **Necessity:** *There exist DAG mixtures for which it is impossible to determine whether $j \in \mathrm{pa}_{\mathrm{m}}(i)$ using CI tests on $p_{\mathrm{m},\mathcal{I}}(x)$ for any intervention $\mathcal{I}$ with $|\mathcal{I}| \leq |\mathrm{pa}_{\mathrm{m}}(i)|$.*

Theorem 1 represents a fundamental step for understanding the intricacies of mixture causal discovery and serves as a guide for evaluating the optimality and efficiency of any learning algorithm. We also note that the necessity statement reflects a worst-case scenario. As such, we present the following refined sufficiency results that can guide efficient algorithm designs.

**Lemma 2.** *Consider nodes $i, j \in \mathbf{V}$ in a mixture of DAGs. It is possible to determine whether $j \in \mathrm{pa}_{\mathrm{m}}(i)$ using CI tests on $p_{\mathrm{m},\mathcal{I}}$ and any of the following interventions:*

*(i)* $\mathcal{I} = \{j\} \cup \bigcup_{\ell \in [K]} \{\mathrm{pa}_{\ell}(i) \cap \mathrm{de}_{\ell}(j)\}$ *; or*

*(ii)* $\mathcal{I} = \{j\} \cup \bigcup_{\ell \in [K]} \{\mathrm{an}_{\ell}(i) \cap \mathrm{ch}_{\ell}(j)\}$ *; or*

*(iii)* $\mathcal{I} = \{j\} \cup \bigcup_{\ell \in [K]} \{\mathrm{an}_{\ell}(i) \cap \mathrm{de}_{\ell}(j) \cap \Delta\}$ *.*

Note that the three interventions in Lemma 2 can coincide when parents of $i$ in a component DAG are also children of $j$ and are in $\Delta$. This case yields the set $\mathcal{I} = \mathrm{pa}_{\mathrm{m}}(i) \cup \{j\}$ with size $(|\mathrm{pa}_{\mathrm{m}}(i)| + 1)$. Since this can be a rare occurrence for realistic mixture models, partial knowledge about the underlying component DAGs, e.g., ancestral relations or the knowledge of $\Delta$, can prove to be useful for identifying $\mathrm{pa}_{\mathrm{m}}(i)$ using interventions with smaller sizes. Finally, we note that our results in Theorem 1 and Lemma 2 are given for a mixture of general DAGs, and they can be improved for special classes of DAGs. In the next result, we focus on mixtures of directed trees.

**Theorem 2** (Intervention sizes – trees). *Consider nodes $i, j \in \mathbf{V}$ in a mixture of $K$ directed trees.*

*(i)* **Sufficiency:** *For any mixture of directed trees, there exists an intervention $\mathcal{I}$ with $|\mathcal{I}| \leq K + 1$ such that it is possible to determine whether $j \in \mathrm{pa}_{\mathrm{m}}(i)$ using CI tests on $p_{\mathrm{m},\mathcal{I}}$.*

*(ii)* **Necessity:** *There exist mixtures of directed trees such that it is impossible to determine whether $j \in \mathrm{pa_m}(i)$ using CI tests on $p_{\mathrm{m},\mathcal{I}}$ for any intervention $\mathcal{I}$ with $|\mathcal{I}| \leq K$.*

Theorem 2 shows that, unlike the general result in Theorem 1, the number of mixture components plays a key role when considering a mixture of directed trees. Hence, prior knowledge of the number of mixture components can be useful for the causal discovery of a mixture of directed trees.

## 4 Learning algorithm and its analysis

In this section, we design an adaptive algorithm that identifies and orients all true edges, referred to as **Ca**usal **D**iscovery from **I**nterventions on **Mixture** Models (CADIM). The algorithm is summarized in Algorithm 1, and its steps are described in Section 4.1. We also analyze the performance guarantees of the algorithm and the optimality of the interventions used in the algorithm in Section 4.2.

### 4.1 Causal discovery from interventions on mixture models

The proposed CADIM algorithm designs interventions for performing causal discovery on a mixture of DAGs. The algorithm is designed to be general and demonstrate feasible time complexity for any mixture of DAGs without imposing structural constraints. Therefore, we forego the computationally expensive task of learning the inseparable pairs from observational data, which requires $\mathcal{O}(n^2 \cdot 2^n)$ CI tests [11], and entirely focus on leveraging interventions for discovering the true causal relationships. The key idea of the algorithm is to use interventions to decompose the ancestors of a node into topological layers and identify the mixture parents by sequentially processing the topological layers using carefully selected interventions. The algorithm consists of four main steps, which are described next.

**Step 1: Identifying mixture ancestors.** We start by identifying the set of mixture ancestors $\mathrm{an_m}(i)$ for each node $i \in \mathbf{V}$, i.e., the union of ancestors of $i$ in the component DAGs. For this purpose, we use single-node interventions. Specifically, for each node $i \in \mathbf{V}$, we intervene on $\mathcal{I} = \{i\}$ and construct the set of nodes that are marginally dependent on $X_i$ in $p_{\mathrm{m},\mathcal{I}}$, i.e.,

$$\hat{\mathrm{de}}(i) = \{j : X_j \not\perp\!\!\!\perp X_i \text{ in } p_{\mathrm{m},\{i\}}\}, \quad \forall i \in \mathbf{V}. \tag{9}$$

Then, we construct the sets $\hat{\mathrm{an}}(i) = \{j : i \in \hat{\mathrm{de}}(j)\}$ for all $i \in \mathbf{V}$. Under $\mathcal{I}$-mixture faithfulness, this procedure ensures that $\hat{\mathrm{de}}(i) = \mathrm{de_m}(i)$, and $\hat{\mathrm{an}}(i) = \mathrm{an_m}(i)$ (see Lemma 3). The rest of the algorithm steps aim to identify mixture parents of a single node $i$, $\mathrm{pa_m}(i)$, within the set $\hat{\mathrm{an}}(i)$. Hence, the following steps can be repeated for all $i \in \mathbf{V}$ to identify all true edges.

**Step 2: Obtaining cycle-free descendants.** In this step, we consider a given node $i \in \mathbf{V}$ and aim to break the *cycles* across the nodes in $\hat{\mathrm{an}}(i)$ by careful interventions. Once this is achieved, for all $j \in \hat{\mathrm{an}}(i)$, we will refine $j$'s descendant set $\hat{\mathrm{de}}(j)$ to *cycle-free* descendant set $\mathrm{de}_i(j)$. The motivation is that these refined descendant sets can be used to topologically order the nodes in $\hat{\mathrm{an}}(i)$. The details of this step work as follows. First, we construct the set of cycles

$$\mathcal{C}(i) \triangleq \{\pi = (\pi_1, \ldots, \pi_\ell) : \pi_1 = \pi_\ell, \forall u \in [\ell-1] \ \pi_u \in \hat{\mathrm{an}}(i) \wedge \pi_u \in \hat{\mathrm{an}}(\pi_{u+1})\}. \tag{10}$$

Subsequently, if $\mathcal{C}(i)$ is not empty, we define a minimal set that shares at least one node with each cycle in $\mathcal{C}(i)$,

$$\mathcal{B}(i) \triangleq \text{ a minimal set such that } \forall \pi \in \mathcal{C}(i) \ |\mathcal{B}(i) \cap \pi| \geq 1. \tag{11}$$

We refer to $\mathcal{B}(i)$ as the *breaking set* of node $i$ since intervening on any set $\mathcal{I}$ that contains $\mathcal{B}(i)$ breaks all the cyclic relationships in $\mathcal{C}(i)$. Then, if $\mathcal{C}(i)$ is not empty, we sequentially intervene on $\mathcal{I} = \mathcal{B}(i) \cup \{j\}$ for all $j \in \hat{\mathrm{an}}(i)$, and construct the cycle-free descendant sets defined as

$$\mathrm{de}_i(j) \leftarrow \{k \in \hat{\mathrm{an}}(i) \cup \{i\} : X_j \not\perp\!\!\!\perp X_k \text{ in } p_{\mathrm{m},\mathcal{I}}\}, \quad \text{where } \mathcal{I} = \mathcal{B}(i) \cup \{j\}. \tag{12}$$

Note that $\mathrm{de}_i(j)$ is a subset of $\mathrm{de_m}(j)$ since intervening on $j$ makes it independent of all its non-descendants. Finally, we construct the set $\mathcal{A} = \{j \in \hat{\mathrm{an}}(i) : i \in \mathrm{de}_i(j)\}$.

**Step 3: Topological layering.** In this step, we decompose $\hat{\mathrm{an}}(i)$ into topological layers by using the cycle-free descendant sets constructed in Step 2. We start by constructing the first layer as

$$S_1(i) = \{j \in \mathcal{A} : \mathrm{de}_i(j) \cap \mathcal{A} = \emptyset\}. \tag{13}$$

---

**Algorithm 1 Ca**usal **D**iscovery from **I**nterventions on **M**ixture Models (CADIM)

---

1: **Step 1: Identify mixture ancestors**
2: **for** $i \in \mathbf{V}$ **do**
3:     Intervene on $\mathcal{I} = \{i\}$, observe samples from $p_{\mathrm{m},\mathcal{I}}$
4:     $\hat{\mathrm{de}}(i) \leftarrow \{j : X_j \not\perp\!\!\!\perp X_i \text{ in } p_{\mathrm{m},\mathcal{I}}\}$                    ▷ mixture descendants of node $i$
5: **for** $i \in \mathbf{V}$ **do**
6:     $\hat{\mathrm{an}}(i) \leftarrow \{j : i \in \hat{\mathrm{de}}(j)\}$                               ▷ mixture ancestors of node $i$

---

7: **Repeat Steps 2, 3, 4 for all** $i \in \mathbf{V}$

---

8: **Step 2: Obtain cycle-free descendants**
9: Find cycles among $\hat{\mathrm{an}}(i)$
10: $\mathcal{C}(i) \leftarrow \{\pi = (\pi_1, \ldots, \pi_{t+1}) \; : \; \pi_1 = \pi_{t+1}, \forall u \in [t] \;\; \pi_u \in \hat{\mathrm{an}}(i) \;\wedge\; \pi_u \in \hat{\mathrm{an}}(\pi_{u+1})\}$
11: **if** $\mathcal{C}(i)$ is empty **then**
12:     $\mathcal{B}(i) \leftarrow \emptyset$
13:     **for** $j \in \hat{\mathrm{an}}(i)$ **do**
14:         $\mathrm{de}_i(j) \leftarrow \{\hat{\mathrm{de}}(j) \cap \hat{\mathrm{an}}(i)\} \cup \{i\}$
15: **else**
16:     $\mathcal{B}(i) \leftarrow$ a minimal set such that $\forall \pi \in \mathcal{C}(i) \;\; |\mathcal{B}(i) \cap \pi| \geq 1$
17:     **for** $j \in \hat{\mathrm{an}}(i)$ **do**
18:         Intervene on $\mathcal{I} = \mathcal{B}(i) \cup \{j\}$                        ▷ break cycles among $\hat{\mathrm{an}}(i)$
19:         $\mathrm{de}_i(j) \leftarrow \{k \in \hat{\mathrm{an}}(i) \cup \{i\} : X_j \not\perp\!\!\!\perp X_k \text{ in } p_{\mathrm{m},\mathcal{I}}\}$   ▷ cycle-free descendants of node $j$
20: $\mathcal{A} \leftarrow \{j \in \hat{\mathrm{an}}(i) \; : \; i \in \mathrm{de}_i(j)\}$                        ▷ refined ancestors

---

21: **Step 3: Topological layering**
22: $t \leftarrow 0$
23: **while** $|\mathcal{A}| \geq 1$ **do**
24:     $t \leftarrow t + 1$
25:     $S_t(i) \leftarrow \{j \in \mathcal{A} : \mathrm{de}_i(j) \cap \mathcal{A} = \emptyset\}$
26:     $\mathcal{A} \leftarrow \mathcal{A} \setminus S_t(i)$

---

27: **Step 4: Identify mixture parents**
28: $\hat{\mathrm{pa}}(i) \leftarrow \emptyset$
29: **for** $u \in (1, \ldots, t)$ **do**
30:     **for** $j \in S_u(i)$ **do**
31:         Intervene on $\mathcal{I} = \hat{\mathrm{pa}}(i) \cup \mathcal{B}(i) \cup \{j\}$
32:         **if** $X_j \not\perp\!\!\!\perp X_i$ in $p_{\mathrm{m},\mathcal{I}}$ **then**
33:             $\hat{\mathrm{pa}}(i) \leftarrow \hat{\mathrm{pa}}(i) \cup \{j\}$
34: **Return** $\hat{\mathrm{pa}}(i)$

---

The construction of cycle-free descendant sets ensures that $S_1(i)$ is not empty. Next, we update $\mathcal{A} \leftarrow \mathcal{A} \setminus S_1(i)$ by removing layer $S_1(i)$ to conclude the first step. Then, we iteratively construct the layers $S_u(i) = \{j \in \mathcal{A} : \hat{\mathrm{de}}(j) \cap \mathcal{A} = \emptyset\}$ and update $\mathcal{A} \leftarrow \mathcal{A} \setminus S_u(i)$ as in Line 26 of the algorithm. We continue until the set $\mathcal{A}$ is exhausted, and denote these topological layers by $\{S_1(i), \ldots, S_t(i)\}$.

**Step 4: Identifying the mixture parents.** Finally, we process the topological layers sequentially to identify the mixture parents in each layer. For a node $j \in S_1(i)$, whether $j \in \mathrm{pa}_{\mathrm{m}}(i)$ can be determined from a marginal independence test on $p_{\mathrm{m},\mathcal{I}}$ where $\mathcal{I} = \mathcal{B}(i) \cup \{j\}$. Leveraging this result, when processing each $S_u(i)$, we consider the nodes $j \in S_u(i)$ sequentially and intervene on $\mathcal{I} = \hat{\mathrm{pa}}(i) \cup \mathcal{B}(i) \cup \{j\}$, where $\hat{\mathrm{pa}}(i)$ denotes the estimated mixture parents. Under this intervention, a statistical dependence implies a true edge from $j$ to $i$. Hence, we update the set $\hat{\mathrm{pa}}(i)$ as follows.

$$\hat{\mathrm{pa}}(i) \leftarrow \hat{\mathrm{pa}}(i) \cup \{j\} \quad \text{if} \;\; X_j \not\perp\!\!\!\perp X_i \text{ in } p_{\mathrm{m},\mathcal{I}} \;\; \text{where} \;\; \mathcal{I} = \hat{\mathrm{pa}}(i) \cup \mathcal{B}(i) \cup \{j\}. \tag{14}$$

After the last layer $S_t(i)$ is processed, the algorithm returns the estimated mixture parents $\hat{\mathrm{pa}}(i)$. By repeating Steps 2, 3, and 4 for all $i \in \mathbf{V}$, we determine the true edges with their orientations.

## 4.2  Guarantees of the CADIM algorithm

In this section, we establish the guarantees of the CADIM algorithm and interpret them vis-à-vis the results in Section 3. We start by providing the following result to show the correctness of identifying mixture ancestors.

**Lemma 3.** *Given $\mathcal{I}$-mixture faithfulness, Step 1 of Algorithm 1 identifies $\{\mathrm{an_m}(i) : i \in [n]\}$ using $n$ single-node interventions.*

Note that the mixture ancestor sets $\{\mathrm{an_m}(i)\}$ do not imply a topological order over the nodes $\mathbf{V}$, e.g., there may exist nodes $u, v$ such that $u \in \mathrm{an_m}(v)$ and $v \in \mathrm{an_m}(u)$. As such, a major difficulty in learning a mixture of DAGs compared to learning a single DAG is the possible cyclic relationships formed by the combination of components of the mixture. Recall that the breaking set is specified in (11) to treat such possible cycles carefully. We refer to the size of $\mathcal{B}(i)$ as the *cyclic complexity number* of node $i$, denoted by $\tau_i$, and the size of the largest breaking set by $\tau_\mathrm{m}$ as

$$\tau_i \triangleq |\mathcal{B}(i)|, \quad \forall i \in \mathbf{V}, \qquad \text{and} \quad \tau_\mathrm{m} \triangleq \max_{i \in \mathbf{V}} \tau_i. \tag{15}$$

Note that $\tau_i$ is readily bounded by the number of cycles in $\mathcal{C}(i)$. Next, we analyze the guarantees of the algorithm for a node $i$ in two cases: $\tau_i = 0$ (cycle-free case) and $\tau_i \geq 1$ (nonzero cyclic complexity).

**Cycle-free case.**  Our next result shows that if $\tau_i = 0$, i.e., there are no cycles among the nodes in $\mathrm{an_m}(i)$, then we identify the mixture parents $\mathrm{pa_m}(i)$, i.e., the union of the nodes that are parents of $i$ in at least one component DAG, using interventions with the optimal size.

**Theorem 3** (Guarantees for cycle-free ancestors). *If the cyclic complexity of node $i$ is zero, then Algorithm 1 ensures that $\hat{\mathrm{pa}}(i) = \mathrm{pa_m}(i)$ by using $|\mathrm{an_m}(i)|$ interventions where the size of each intervention is at most $|\mathrm{pa_m}(i)| + 1$.*

Theorem 3 shows that by repeating the algorithm steps for each node $i \in \mathbf{V}$, we can identify all true edges with their orientations using $n + \sum_{i \in \mathbf{V}} |\mathrm{an_m}(i)| \leq n + n(n-1) = n^2$ interventions, where the size of each intervention is bounded by the worst-case necessary size established in Theorem 1.

**Nonzero cyclic complexity.**  Finally, we address the most general case, in which the mixture ancestors of node $i$ might contain cycles. In this case, our algorithm performs additional interventions to break the cycles among $\mathrm{an_m}(i)$. Hence, the number and size of the interventions will be greater than the cycle-free case, which is established in the following result.

**Theorem 4** (Guarantees for general mixtures). *Algorithm 1 ensures that $\hat{\mathrm{pa}}(i) = \mathrm{pa_m}(i)$ by using $|\mathrm{an_m}(i)|$ interventions with size $\tau_i + 1$, and $|\mathrm{an_m}(i)|$ interventions with size at most $|\mathrm{pa_m}(i)| + \tau_i + 1$.*

Theorem 4 shows that, Algorithm 1 achieves the causal discovery objectives by using a total of $n + 2 \sum_{i \in \mathbf{V}} |\mathrm{an_m}(i)| \leq n + 2n(n-1) = \mathcal{O}(n^2)$ interventions, where the maximum intervention size for learning each $\mathrm{pa_m}(i)$ is at most $\tau_i$ larger than the necessary and sufficient size $|\mathrm{pa_m}(i)| + 1$. This optimality gap reflects the challenges of accommodating cyclic relationships in intervention design for learning in mixtures while also maintaining a quadratic number of interventions $\mathcal{O}(n^2)$.

## 5  Experiments

We evaluate the performance of Algorithm 1 for estimating the true edges in a mixture of DAGs using synthetic data and investigate the need for interventions, the effect of the graph size, and the cyclic complexity. Additional results for varying the number of components, parameterization, and number of samples are provided in Appendix E [1].

**Experimental setup.** We use an Erdős-Rényi model $G(n, p)$ with density $p = 2/n$ to generate the component DAGs $\{\mathcal{G}_\ell : \ell \in [K]\}$ for different values of nodes $n$ and mixture components $K$. We adopt linear structural equation models (SEMs) with Gaussian noise for the causal models, in which the noise for node $i$ is sampled from $\mathcal{N}(\mu_i, \sigma_i^2)$ where $\mu_i$ is sampled uniformly in $[-1, 1]$ and $\sigma_i^2$ is sampled uniformly in $[0.5, 1.5]$. The edge weights are sampled uniformly in $\pm[0.25, 2]$. We

---

[1]The codebase for the experiments can be found at `https://github.com/bvarici/intervention-mixture-DAG`.

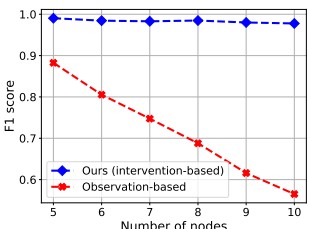 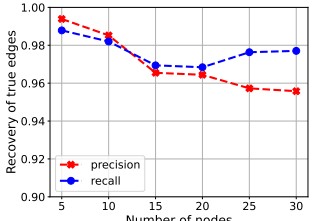 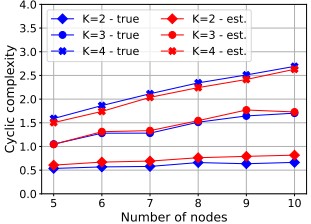

(a) Skeleton recovery for observa-
tional and interventional methods

(b) Varying the number of nodes
$n$ for a mixture of $K = 3$ DAGs

(c) Empirical cyclic complexity
for varying $n$ and $K$ values

Figure 2: Mean true edge recovery rates and quantification of mean cyclic complexity of a node.

consider the case where a change in the conditional distribution of node $i$ is only caused by changes in the parents of $i$ across different DAGs. We use a partial correlation test to check (conditional) independence in the algorithm steps, similar to the related work [10, 11]. We repeat this procedure for 100 randomly generated DAG mixtures for each of the following settings.

**Need for interventions.** We demonstrate the need for interventions for learning the skeleton in the mixture of DAGs, unlike the case of single DAGs. To this end, we consider a mixture of $K = 2$ DAGs and learn the inseparable node pairs via exhaustive CI tests (see Algorithm 2 in Appendix E). Figure 2a empirically verifies the claim that true edges (even their undirected versions) cannot be learned using observational data only.

**Recovery of true edges.** We evaluate the performance of Algorithm 1 on the central task of learning the true edges in the mixture. For this purpose, we report average precision and recall rates for recovering the true edges. We look into the performance of Algorithm 1 under a varying number of nodes $n \in [5, 30]$ for a mixture of $K = 3$ DAGs and using 5000 samples from each DAG. Figure 2b demonstrates that Algorithm 1 maintains a strong performance even under $n = 30$ nodes. We provide additional results for the number of DAGs in the range $K \in [2, 10]$ and varying number of samples in Appendix E.

**Quantification of cyclic complexity.** We recall that for finding the mixture parents of a node $i$, the maximum size of the intervention used in Algorithm 1 is at most $\tau_i$, i.e., cyclic complexity, larger than the necessary size. In Figure 2c, we plot the empirical values of average cyclic complexity – both the ground truth and estimated by the algorithm. Figure 2c shows that even though average $\tau_i$ increases with $K$, it still remains very small, e.g., approximately 1.5 for a mixture of $K = 3$ DAGs with $n = 10$ nodes. Furthermore, on average, the estimated $\tau_i$ values used in the algorithm are almost identical to the ground truth $\tau_i$. Therefore, Algorithm 1 maintains its close to optimal intervention size guarantees in the finite-sample regime.

## 6  Conclusion

In this paper, we have conducted the first analysis of using interventions to learn causal relationships in a mixture of DAGs. First, we have established the matching necessary and sufficient size of interventions needed for learning the true edges in a mixture. Subsequently, guided by this result, we have designed an algorithm that learns the true edges using interventions with close to optimal sizes. We have also analyzed the optimality gap of our algorithm in terms of the cyclic relationships within the mixture model. The proposed algorithm uses a total of $\mathcal{O}(n^2)$ interventions. Establishing lower bounds for the number of interventions with constrained sizes remains an important direction for future work, which can draw connections to intervention design for single-DAG and further characterize the differences of causal discovery in mixtures. Finally, generalizing the mixture model to accommodate partial knowledge of the underlying domains can be useful in disciplines where such knowledge can be acquired a priori.

## Acknowledgments and disclosure of funding

This work was supported by IBM through the IBM-Rensselaer Future of Computing Research Collaboration.

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

# Interventional Causal Discovery in a Mixture of DAGs
# Appendices

## Table of Contents

## A  Auxiliary results

**Lemma 4** (Markov property, [10, Theorem 3.2]). *Let $A, B, C \subseteq \mathbf{V}$ be disjoint. If $\bar{A}$ and $\bar{B}$ are d-separated given $\bar{C}$ in the mixture DAG, then $X_A$ and $X_B$ are conditionally independent given $X_C$ in mixture distribution.*

**Lemma 5** ([11, Theorem 5]). *Consider nodes $i, j \in \mathbf{V}$ such that $i$ and $j$ are not adjacent in any of the component DAGs, i.e., $i \notin \mathrm{pa_m}(j)$ and $j \notin \mathrm{pa_m}(i)$.*

*(i) If $i \in \Delta$ and $j \in \Delta$: $i$ and $j$ are always inseparable, i.e., $(i - j)$ is an emergent edge.*

*(ii) If $i \notin \Delta$ and $j \notin \Delta$: If $i$ and $j$ are inseparable, then there exist two component DAGs $\mathcal{G}_\ell$, $\mathcal{G}_{\ell'}$ such that $\mathcal{G}_\ell$ contains a $\Delta$-through path from $i$ to $j$ and $\mathcal{G}_{\ell'}$ contains a $\Delta$-through path from $j$ to $i$.*

*(iii) If $i \notin \Delta$ and $j \in \Delta$: If $i$ and $j$ are inseparable, then at least one component DAG contains a $\Delta$-through path from $i$ to $j$.*

**Lemma 6** ([11, Theorem 6]). *Suppose that $\mathcal{G}_1, \ldots, \mathcal{G}_K$ are directed trees. Consider nodes $i, j \in \mathbf{V}$ such that $i$ and $j$ are not adjacent in any component DAG, i.e., $i \notin \mathrm{pa_m}(j)$ and $j \notin \mathrm{pa_m}(i)$.*

*(i) If $i \in \Delta$ and $j \in \Delta$: $i$ and $j$ are always inseparable.*

*(ii) If $i \notin \Delta$ and $j \notin \Delta$: $i$ and $j$ are separable if and only if there does not exist $\mathcal{G}_\ell$, $\mathcal{G}_{\ell'}$ such that the two DAGs contain $\Delta$-child-through paths between $i$ and $j$ in opposite directions.*

*(iii) If $i \notin \Delta$ and $j \in \Delta$: $i$ and $j$ are separable if and only if none of the component DAGs contains a $\Delta$-child-through path from $i$ to $j$.*

# B Proofs for Section 3

## B.1 Proof of Lemma 1

**Proof of identifiability:** Since $j \in \mathcal{I}$, we have $j \notin \Delta_{\mathcal{I}}$. Then, Lemma 5 (iii) implies that if $(j \to i)$ is not a true edge and there does not exist a $\Delta$-through path from $j$ to $i$ in any $\mathcal{G}_{\ell,\mathcal{I}}$, then $i$ and $j$ are separable in $p_{\mathrm{m},\mathcal{I}}$. Consequently, whether $j \in \mathrm{pa}_{\mathrm{m}}(i)$ can be determined from $p_{\mathrm{m},\mathcal{I}}$.

**Proof of non-identifiability:** First, note that using Lemma 4, for any intervention $\mathcal{I}$, $p_{\mathrm{m},\mathcal{I}}$ is Markov with respect to the $\mathcal{I}$-mixture DAG $\mathcal{G}_{\mathrm{m},\mathcal{I}}$. Then, for $j \in \Delta_{\mathcal{I}}$, if also $i \in \Delta_{\mathcal{I}}$, $i$ and $j$ are inseparable in $p_{\mathrm{m},\mathcal{I}}$ regardless of whether there is a true edge between $i$ and $j$. Hence, whether $j \in \mathrm{pa}_{\mathrm{m}}(i)$ cannot be determined using CI tests on $p_{\mathrm{m},\mathcal{I}}$. For the second statement, recall that intervention $\mathcal{I}$ cuts off all incoming edges to nodes of $\mathcal{I}$ in all component DAGs. Then, if $i \in \mathcal{I}$, we cannot determine whether $j \in \mathrm{pa}_{\mathrm{m}}(i)$ since the possible influence of $j$ on $i$ is cut off by the intervention. Suppose that $i \in \Delta_{\mathcal{I}}$, and let $\pi$ be a $\Delta$-child-through path from $j$ to $i$ in some component DAG $\mathcal{G}_{\ell,\mathcal{I}}$, i.e., $\pi$ is given by $j \xrightarrow{\ell} k \overset{\ell}{\rightsquigarrow} i$ for some $k \in \Delta_{\mathcal{I}}$. Since $i \in \Delta_{\mathcal{I}}$, the $\mathcal{I}$-mixture DAG $\mathcal{G}_{\mathrm{m},\mathcal{I}}$ also contains the path $j \xrightarrow{\ell} k \xleftarrow{\ell} y \xrightarrow{\ell} i$. Since these two paths cannot be blocked simultaneously by conditioning on a set of nodes, $i$ and $j$ are inseparable regardless of whether $j \in \mathrm{pa}_{\mathrm{m}}(i)$. Therefore, whether $j \in \mathrm{pa}_{\mathrm{m}}(i)$ cannot be determined using CI tests on $p_{\mathrm{m},\mathcal{I}}$.

## B.2 Proof of Lemma 2

We start by providing a general statement that will be used for the proof of the three subcases. Let $\mathcal{I}$ be an intervention such that $j \in \mathcal{I}$ and there does not exist a $\Delta$-through path from $j$ to $i$ in any component DAG $\mathcal{G}_{\ell,\mathcal{I}}$. In this case, using Lemma 5, if $(j \to i)$ is not a true edge, then $i$ and $j$ are separable in $p_{\mathrm{m},\mathcal{I}}$. Subsequently, if $i$ and $j$ are inseparable in $p_{\mathrm{m},\mathcal{I}}$, then $(j \to i)$ is a true edge.

Let $\pi$ be a $\Delta$-through path from $j$ to $i$ in some $\mathcal{G}_{\ell,\mathcal{I}}$. Note that, for any of the following three intervention sets,

(i) $\mathcal{I} = \{j\} \cup \bigcup_{\ell \in [K]} \left\{ \mathrm{pa}_{\ell}(i) \cap \mathrm{de}_{\ell}(j) \right\}$ ,

(ii) $\mathcal{I} = \{j\} \cup \bigcup_{\ell \in [K]} \left\{ \mathrm{an}_{\ell}(i) \cap \mathrm{ch}_{\ell}(j) \right\}$ ,

(iii) $\mathcal{I} = \{j\} \cup \bigcup_{\ell \in [K]} \left\{ \mathrm{an}_{\ell}(i) \cap \mathrm{de}_{\ell}(j) \cap \Delta \right\}$ ,

$\pi$ cannot contain a node between $j$ and $i$. Therefore, if $j \notin \mathrm{pa}_{\mathrm{m}}(i)$, there does not exist a $\Delta$-through path from $j$ to $i$. Subsequently, if $i$ and $j$ are not separable in $p_{\mathrm{m},\mathcal{I}}$ for any of these three interventions, it means there exists $j \xrightarrow{\ell} i$ for some component DAG $\mathcal{G}_{\ell}$. Therefore, whether $j \in \mathrm{pa}_{\mathrm{m}}(i)$ can be determined by checking whether $i$ and $j$ are separable in $p_{\mathrm{m},\mathcal{I}}$ for any of these three interventions.

## B.3 Proof of Theorem 1

The sufficiency result immediately follows from Lemma 2 since each of the three interventions in stated in Lemma 2 is a subset of $\mathrm{pa}_{\mathrm{m}}(i) \cup \{j\}$. To show the worst-case necessity of an intervention with size at least $|\mathrm{pa}_{\mathrm{m}}(i)| + 1$, we construct the following example. Consider component DAGs $\{\mathcal{G}_{\ell} : \ell \in [K]\}$ such that $\mathcal{G}_1$ contains a single edge $i \xrightarrow{1} j$. In the rest of the component DAGs, for any $k \xrightarrow{\ell} i$ edge, let us also draw $j \xrightarrow{\ell} k$. We do not put any constraints on the other possible connections in $\{\mathcal{G}_{\ell} : \ell \in \{2, \ldots, K\}\}$. Note that this construction yields that $\mathrm{pa}_{\mathrm{m}}(i) \cup \{i, j\} \subseteq \Delta$. Consider the paths

$$j \xleftarrow{1} y \xrightarrow{1} i , \quad \text{and} \quad \{j \xrightarrow{\ell} k \xrightarrow{\ell} i : k \in \mathrm{pa}_{\mathrm{m}}(i)\} . \tag{16}$$

For any intervention $\mathcal{I} \subseteq \mathbf{V} \setminus \{i\}$ that does not contain all nodes in $\mathrm{pa}_{\mathrm{m}}(i) \cup \{j\}$, at least one of these paths will be active in the $\mathcal{I}$-mixture DAG $\mathcal{G}_{\mathrm{m},\mathcal{I}}$, regardless of whether there exists a $j \xrightarrow{\ell} i$, $\ell \in \{2, \ldots, K\}$ edge. Therefore, at the worst-case, whether $j \in \mathrm{pa}_{\mathrm{m}}(i)$ cannot be determined from $p_{\mathrm{m},\mathcal{I}}$ for any intervention $\mathcal{I}$ with size $|\mathcal{I}| \leq |\mathrm{pa}_{\mathrm{m}}(i)|$.

### B.4 Proof of Theorem 2

**Proof of sufficiency.** Consider an inseparable pair $(i - j)$ in $p_\mathrm{m}$. For a mixture of directed trees, Lemma 6 implies that if there does not exist a $\Delta$-child-through path from $j$ to $i$ in any component DAG, then $(i - j)$ corresponds to a true edge. Consider the intervention

$$\mathcal{I} = \{j\} \cup \bigcup_{\ell \in [K]} \{\mathrm{an}_\ell(i) \cap \mathrm{ch}_\ell(j) \cap \Delta\}, \tag{17}$$

which cuts off all $\Delta$-child-through paths from $j$ to $i$ in component DAGs. Also note that $|\mathrm{an}_\ell(i) \cap \mathrm{ch}_\ell(j) \cap \Delta| \le 1$ since each $\mathcal{G}_\ell$ contains at most one causal path from $j$ to $i$. Then, $i$ and $j$ are inseparable in $p_{\mathrm{m},\mathcal{I}}$ if and only if $j \in \mathrm{pa}_\mathrm{m}(i)$, since $j \in \mathcal{I}$ cuts off all $i \xrightarrow{\ell} j$ edges. Therefore, we can determine whether $j \in \mathrm{pa}_\mathrm{m}(i)$ from $p_{\mathrm{m},\mathcal{I}}$ with an intervention $\mathcal{I}$ where

$$|\mathcal{I}| = 1 + \sum_{\ell \in [K]} \mathbb{1}\big(\{\mathrm{an}_\ell(i) \cap \mathrm{ch}_\ell(j) \cap \Delta\} \ne \emptyset\big) \le K + 1. \tag{18}$$

**Proof of necessity.** For the worst-case necessity, consider component trees $\{\mathcal{G}_\ell : \ell \in [K]\}$ such that each graph contains a $\Delta$-child-through path from $j$ to $i$ in which the children of $j$ in each graph is distinct. Also, let $k \in \mathrm{pa}_1(j)$ but $k \notin \mathrm{pa}_\ell(j)$ for all $\ell \in \{2, \ldots, K\}$. This construction yields that

$$\mathcal{J} \triangleq \{i, j\} \cup \bigcup_{\ell \in [K]} \{\mathrm{an}_\ell(i) \cap \mathrm{ch}_\ell(j) \cap \Delta\} \subseteq \Delta. \tag{19}$$

Consider the paths

$$j \xleftarrow{1} y \xrightarrow{1} i, \quad \text{and} \quad \{j \xrightarrow{\ell} k \overset{\ell}{\rightsquigarrow} i : k \in \mathcal{J} \setminus \{i\}\}. \tag{20}$$

For any intervention $\mathcal{I} \subseteq \mathbf{V} \setminus \{i\}$ that does not contain all nodes in $\mathcal{J} \setminus \{i\}$, at least one of these paths will be active in the $\mathcal{I}$-mixture DAG $\mathcal{G}_{\mathrm{m},\mathcal{I}}$, regardless of whether one of the $\Delta$-child-through paths is $j \xrightarrow{\ell} i$ itself. Note that if $j \notin \mathrm{pa}_\mathrm{m}(i)$, then this specific construction yields that $|\mathcal{J} \setminus \{i\}| = K + 1$. Therefore, at the worst-case, whether $j \in \mathrm{pa}_\mathrm{m}(i)$ cannot be determined from $p_{\mathrm{m},\mathcal{I}}$ for any intervention $\mathcal{I}$ with size $|\mathcal{I}| \le K$.

## C  Proofs for Section 4

### C.1  Proof of Lemma 3

Consider intervention $\mathcal{I} = \{i\}$ and the corresponding interventional mixture distribution $p_{\mathrm{m},\mathcal{I}}$ and $\mathcal{I}$-mixture DAG $\mathcal{G}_{\mathrm{m},\mathcal{I}}$. We will show that $\{j : X_j \not\perp\!\!\!\perp X_i \text{ in } p_{\mathrm{m},\mathcal{I}}\}$ is equal to the mixture descendants $\mathrm{de}_\mathrm{m}(i)$. First, let $i \in \mathrm{an}_\mathrm{m}(j)$, i.e., there exists a path $i \overset{\ell}{\rightsquigarrow} j$ for some $\ell \in [K]$. Then, by $\mathcal{I}$-mixture faithfulness, $X_j \not\perp\!\!\!\perp X_i$ in $p_{\mathrm{m},\mathcal{I}}$. For the other direction, let $X_j \not\perp\!\!\!\perp X_i$ in $p_{\mathrm{m},\mathcal{I}}$. Then, there must be an active path between $j$ and $i$ in $\mathcal{I}$-mixture DAG $\mathcal{G}_{\mathrm{m},\mathcal{I}}$. Since $i$ is intervened and there is no conditioning set, the path cannot contain any collider, which implies that the path is in the form $i \overset{\ell}{\rightsquigarrow} j$ for some $\ell \in [K]$. Therefore, we have

$$i \in \mathrm{an}_\mathrm{m}(j) \quad \Longleftrightarrow \quad X_j \not\perp\!\!\!\perp X_i \text{ in } p_{\mathrm{m},\{i\}}, \tag{21}$$

which concludes the proof.

**The choice of single-node interventions in Step 1.** We note that the choice of single-node interventions for learning the mixture ancestors $\{\mathrm{an}_\mathrm{m}(i) : i \in \mathbf{V}\}$ is deliberate for simplicity. It is possible to achieve the same guarantee using fewer than $n$ interventions by designing multi-node interventions, e.g., using separating systems with restricted intervention sizes [16]. Nevertheless, using $n$ intervention does not compromise our main results since as we show in the sequel, the number of total interventions in the algorithm will be dominated by the number of interventions in the subsequent steps.

### C.2  Proof of Theorem 3

Lemma 3 ensures that Step 1 of Algorithm 1 identifies $\mathrm{an}_\mathrm{m}(i)$ and $\mathrm{de}_\mathrm{m}(i)$ correctly for all $i \in [n]$. Hence, in this proof we use $\mathrm{an}_\mathrm{m}(i)$ and $\mathrm{de}_\mathrm{m}(i)$ for $\hat{\mathrm{an}}(i)$ and $\hat{\mathrm{de}}(i)$, respectively. We consider the

case where there are no cycles among the mixture ancestors of node $i$ induced by the mixture ancestral relationships, i.e., the following set of cycles is empty.

$$\mathcal{C}(i) \leftarrow \{\pi = (\pi_1, \ldots, \pi_\ell) \ : \ \pi_1 = \pi_\ell \,, \forall u \in [\ell - 1] \ \ \pi_u \in \mathrm{an_m}(i) \ \wedge \ \pi_u \in \mathrm{an_m}(\pi_{u+1})\} \,. \quad (22)$$

In this case, Step 2 of Algorithm 1 only creates the sets

$$\mathrm{de}_i(j) \triangleq \hat{\mathrm{de}}(j) \cap \{\hat{\mathrm{an}}(i) \cup i\} = \mathrm{de_m}(j) \cap \{\mathrm{an_m}(i) \cup i\} \,, \quad \forall i \in \mathrm{an_m}(i) \,, \quad (23)$$

and $\mathcal{A} = \mathrm{an_m}(i)$. The lack of cycles implies that the nodes in $\mathcal{A}$ can be topologically ordered, i.e., there exists an ordering $(\mathcal{A}_1, \ldots, \mathcal{A}_{|\mathrm{an_m}(i)|})$ such that $\mathcal{A}_j \in \mathrm{de_m}(\mathcal{A}_k)$ implies $j < k$. In Step 3, we leverage this key property for constructing hierarchically ordered topological layers.

Next, recall the definition of $S_1(i)$ in (13),

$$S_1(i) = \{j \in \mathrm{an_m}(i) : \mathrm{de_m}(j) \cap \mathrm{an_m}(i) = \emptyset\} \,. \quad (24)$$

Then, since there are no cycles among the nodes in $\mathrm{an_m}(i)$, $S_1(i)$ is not empty, e.g., the first node of the topological order described above has no mixture descendant within $\mathrm{an_m}(i)$. Consider $j \in S_1(i)$. Since $\mathrm{an_m}(i) \cap \mathrm{de_m}(j) = \emptyset$, $j$ must be a mixture parent of $i$. Therefore, $S_1(i) \subseteq \mathrm{pa_m}(i)$. We will use induction to prove that topological layering in Step 3 and the sequential interventions in Step 4 ensure identifying $\mathrm{pa_m}(i)$.

**Base case.** Consider $S_2(i)$ defined as

$$S_2(i) = \{j \in \mathrm{an_m}(i) \setminus S_1(i) : \mathrm{de_m}(j) \cap \{\mathrm{an_m}(i) \setminus S_1(i)\} = \emptyset\} \,. \quad (25)$$

Note that we have $\hat{\mathrm{pa}}(i) = S_1(i)$. Consider a node $j \in S_2(i)$ and intervene on $\mathcal{I} = S_1(i) \cup \{j\}$. If $j \in \mathrm{pa_m}(i)$, then by $\mathcal{I}$-mixture faithfulness, $X_j \not\perp\!\!\!\perp X_i$ in $p_{\mathrm{m}, \mathcal{I}}$. On the other direction, suppose that $X_j \not\perp\!\!\!\perp X_i$ in $p_{\mathrm{m}, \mathcal{I}}$. If $j \notin \mathrm{pa_m}(i)$, then there exists an active path between $\bar{j}$ and $\bar{i}$ in $\mathcal{G}_{\mathrm{m}, \mathcal{I}}$. Since the conditioning set is empty, there cannot be any colliders on the path. Then, since $j \notin \mathrm{pa_m}(i)$, the path has the form $j \overset{\ell}{\rightsquigarrow} k \overset{\ell}{\rightarrow} i$ for some $k \in \mathrm{an_m}(i)$ and $\ell \in [K]$. We know that $k \in S_1(i)$ since intervention $\mathcal{I}$ contains $S_1(i)$. Then, $k \in \mathrm{de_m}(j) \cap \{\mathrm{an_m}(i) \setminus S_1(i)\}$, which contradicts with $j \in S_2(i)$ by definition of $S_2(i)$. Therefore, $X_j \not\perp\!\!\!\perp X_i$ in $p_{\mathrm{m}, \mathcal{I}}$ implies that $j \in \mathrm{pa_m}(i)$. Subsequently, for $j \in S_2(i)$, we have

$$j \in \mathrm{pa_m}(i) \quad \Longleftrightarrow \quad X_j \not\perp\!\!\!\perp X_i \ \text{in} \ p_{\mathrm{m}, \mathcal{I}} \,. \quad (26)$$

Note that this step uses $|S_2(i)|$ interventions, one for each $j \in S_2(i)$, and each intervention has size $|\mathcal{I}| = |S_1(i)| + 1$.

**Induction hypothesis.** Assume that we have identified the set $S_u(i) \cap \mathrm{pa_m}(i)$ correctly for $u \in \{1, \ldots, v - 1\}$, i.e., we have $\hat{\mathrm{pa}}(i) = \bigcup_{k=1}^{v-1} S_k(i) \cap \mathrm{pa_m}(i)$. We will show that the algorithm also identifies $S_v(i) \cap \mathrm{pa_m}(i)$ correctly.

Let $\mathcal{A} = \mathrm{an_m}(i) \setminus \bigcup_{k=1}^{v-1} S_k(i)$ and consider $S_v(i)$ defined as

$$S_v(i) = \{j \in \mathcal{A} : \mathrm{de_m}(j) \cap \mathcal{A} = \emptyset\} \,. \quad (27)$$

Consider a node $j \in S_v(i)$ and intervene on

$$\mathcal{I} = \{j\} \cup \hat{\mathrm{pa}}(i) = \{j\} \cup \bigcup_{k=1}^{v-1} S_k(i) \cap \mathrm{pa_m}(i) \,. \quad (28)$$

If $j \in \mathrm{pa_m}(i)$, then by $\mathcal{I}$-mixture faithfulness, $X_j \not\perp\!\!\!\perp X_i$ in $p_{\mathrm{m}, \mathcal{I}}$. For the other direction, suppose that $X_j \not\perp\!\!\!\perp X_i$ in $p_{\mathrm{m}, \mathcal{I}}$. If $j \notin \mathrm{pa_m}(i)$, then there exists an active path between $\bar{j}$ and $\bar{i}$ in $\mathcal{I}$-mixture DAG $\mathcal{G}_{\mathrm{m}, \mathcal{I}}$. Since the conditioning set is empty, this path has the form $j \overset{\ell}{\rightsquigarrow} k \overset{\ell}{\rightarrow} i$ for some $k \in \mathrm{de_m}(j) \cap \mathcal{A}$, which contradicts with $j \in S_v(i)$ by the definition of $S_v(i)$. Subsequently, for $j \in S_v(i)$, we have

$$j \in \mathrm{pa_m}(i) \quad \Longleftrightarrow \quad X_j \not\perp\!\!\!\perp X_i \ \text{in} \ p_{\mathrm{m}, \mathcal{I}} \,. \quad (29)$$

Therefore, by induction, the algorithm identifies $S_u(i) \cap \mathrm{pa_m}(i)$ correctly for all $u \in \{1, \ldots, t\}$.

Finally, note that while processing each layer $S_u(i)$, the algorithm uses $|S_u(i)|$ interventions, one for each $j \in S_u(i)$, with size $|\mathcal{I}| = \left|\mathrm{pa_m}(i) \cap \bigcup_{k=1}^{u-1} S_k(i)\right| + 1$. This is upper bounded by $|\mathrm{pa_m}(i)| + 1$, which is shown to be the worst-case necessary intervention size in Theorem 1. Then, including $n$ single-node interventions performed in Step 1, for identifying $\mathrm{pa_m}(i)$ for all $i \in \mathbf{V}$, Algorithm 1 uses a total of

$$n + \sum_{i=1}^{n} \sum_{u=1}^{t} |S_u(i)| = n + \sum_{i=1}^{n} |\mathrm{an_m}(i)| = \mathcal{O}(n^2) \tag{30}$$

interventions, which completes the proof of the theorem.

### C.3  Proof of Theorem 4

We start by giving a synopsis of the proof. Lemma 3 ensures that Step 1 of Algorithm 1 identifies $\mathrm{an_m}(i)$ and $\mathrm{de_m}(i)$ correctly for all $i \in \mathbf{V}$. Hence, in this proof we use $\mathrm{an_m}(i)$ and $\mathrm{de_m}(i)$ for $\hat{\mathrm{an}}(i)$ and $\hat{\mathrm{de}}_\mathrm{m}(i)$, respectively. In this theorem, we consider the most general case in which the nodes in mixture ancestors $\mathrm{an_m}(i)$ can form cycles via their mixture ancestral relationships. These cycles will be accommodated by the procedure in Step 2. Intuitively, by intervening on a small number of nodes, we can break all the cycles in $\mathcal{C}(i)$ in the new interventional mixture graphs. Then, we would be able to follow Steps 3 and 4 similarly to the proof of Theorem 3, albeit using interventions with larger sizes.

**Step 2.**  First, we recall the definition of cycles among mixture ancestors of $i$,

$$\mathcal{C}(i) \leftarrow \{\pi = (\pi_1, \ldots, \pi_\ell) \; : \; \pi_1 = \pi_\ell \,, \forall u \in [\ell - 1] \;\; \pi_u \in \hat{\mathrm{an}}(i) \; \wedge \; \pi_u \in \hat{\mathrm{an}}(\pi_{u+1}) \} \,, \tag{31}$$

and the associated breaking set,

$$\mathcal{B}(i) \triangleq \text{a minimal set s.t. } \forall \pi \in \mathcal{C}(i), \;\; |\mathcal{B}(i) \cap \pi| \geq 1 \,. \tag{32}$$

We denote the size of the breaking set by $\tau_i \triangleq |\mathcal{B}(i)|$ and refer to it as the *cyclic complexity* of node $i$. The intervention $\mathcal{I} = \mathcal{B}(i)$ breaks all cycles in $\mathcal{C}(i)$. To see this consider a cycle $\pi = (\pi_1, \ldots, \pi_\ell)$ in $\mathcal{C}(i)$ and suppose that $pi_u \in \mathcal{B}(i)$. Then, intervening on $\pi_u$ breaks all causal paths from $\pi_{u-1}$ to $\pi_u$, which breaks the cycle. In Step 2, we leverage this property to obtain *cycle-free* descendants of each node $j \in \mathrm{an_m}(i)$. Specifically, for each each $j \in \mathrm{an_m}(i)$, we intervene on $\mathcal{I} = \mathcal{B}(i) \cup \{j\}$ and set

$$\mathrm{de}_i(j) = \{k \in \mathrm{an_m}(i) \cup \{i\} : X_j \not\perp\!\!\!\perp X_k \; \text{in} \; p_{\mathrm{m},\mathcal{I}} \} \,. \tag{33}$$

Note that if $j \in \mathrm{pa_m}(i)$, then $i \in \mathrm{de}_i(j)$. Hence, after constructing these cycle-free descendant sets, we refine the ancestor set

$$\mathcal{A} = \{j \in \mathrm{an_m}(i) : i \in \mathrm{de}_i(j)\} \,, \tag{34}$$

which contains all $\mathrm{pa_m}(i)$. We will use induction to prove that topological layering in Step 3 and the sequential interventions on Step 4 ensure to identify $\mathrm{pa_m}(i)$ from $\mathcal{A}$.

**Base case.**  Consider $S_1(i)$ defined as

$$S_1(i) = \{j \in \mathcal{A} : \mathrm{de}_i(j) \cap \mathcal{A} = \emptyset \} \,. \tag{35}$$

First, we show that $S_1(i)$ is not empty. Otherwise, starting from a node $\pi_1 \in \mathcal{A}$, we would have

$$\mathrm{de}_i(\pi_1) \cap \mathcal{A} \neq \emptyset \quad \implies \quad \exists \, \pi_2 \in \mathrm{de}_i(\pi_1) \cap \mathcal{A} \tag{36}$$

$$\mathrm{de}_i(\pi_2) \cap \mathcal{A} \neq \emptyset \quad \implies \quad \exists \, \pi_3 \in \mathrm{de}_i(\pi_2) \cap \mathcal{A} \tag{37}$$

$$\vdots \tag{38}$$

$$\mathrm{de}_i(\pi_\ell) \cap \mathcal{A} \neq \emptyset \quad \implies \quad \exists \, \pi_1 \in \mathrm{de}_i(\pi_\ell) \cap \mathcal{A} \tag{39}$$

since $\mathcal{A}$ has finite elements. However, this implies that none of the $\{\pi_1, \ldots, \pi_\ell\}$ are contained in $\mathcal{B}(i)$ due to the construction of $\mathrm{de}_i(j)$ sets with interventions $\mathcal{I} = \mathcal{B}(i) \cup \{j\}$. This contradicts with the definition of the breaking set $\mathcal{B}(i)$ as it does not contain any node from the cycle $\{\pi_1, \ldots, \pi_\ell\}$.

Next, consider a node $j \in S_1(i)$ and intervene on $\mathcal{I} = \mathcal{B}(i) \cup \{j\}$. We will show that

$$j \in \mathrm{pa_m}(i) \quad \Longleftrightarrow \quad X_j \not\!\perp\!\!\!\perp X_i \text{ in } p_{\mathrm{m},\mathcal{I}} \,. \tag{40}$$

If $j \in \mathrm{pa_m}(i)$, then by $\mathcal{I}$-mixture faithfulness, $X_j \not\!\perp\!\!\!\perp X_i$ in $p_{\mathrm{m},\mathcal{I}}$. We prove the other direction, that is $X_j \not\!\perp\!\!\!\perp X_i$ in $p_{\mathrm{m},\mathcal{I}}$ implies that $j \in \mathrm{pa_m}(i)$ as follows. First, note that $X_j \not\!\perp\!\!\!\perp X_i$ in $p_{\mathrm{m},\mathcal{I}}$ does not have a conditioning set. Then, it implies that there exists an active path $j \overset{\ell}{\rightsquigarrow} i$ in $\mathcal{G}_{\ell,\mathcal{I}}$ for some $\ell \in [K]$. Suppose that $j \notin \mathrm{pa_m}(i)$, which implies that $j \overset{\ell}{\rightarrow} k \overset{\ell}{\rightsquigarrow} i$ in $\mathcal{G}_{\ell,\mathcal{I}}$, and $k \notin \mathcal{B}(i)$ for path being active. However, in this case we have $k \in \mathrm{de}_i(j) \cap \mathcal{A}$, which contradicts with $j \in S_1(i)$ due to definition of $S_1(i)$. Hence, for $j \in S_1(i)$, $X_j \not\!\perp\!\!\!\perp X_i$ in $p_{\mathrm{m},\mathcal{I}}$ implies that $j \in \mathrm{pa_m}(i)$, which concludes the proof of the base case.

**Induction step.** Assume that we have identified the set $S_u(i) \cap \mathrm{pa_m}(i)$ correctly for $u \in \{1,\ldots,v-1\}$. Let $\mathcal{A} = \mathrm{an_m}(i) \setminus \bigcup_{k=1}^{v-1} S_k(i)$ and consider $S_v(i)$ defined as

$$S_v(i) = \{j \in \mathcal{A} : \mathrm{de}_i(j) \cap \mathcal{A} = \emptyset\} \,. \tag{41}$$

Note that, after processing $\{S_1,\ldots,S_{v-1}\}$ correctly, we have

$$\hat{\mathrm{pa}}(i) = \mathrm{pa_m}(i) \cap \bigcup_{k=1}^{v-1} S_k(i) \,. \tag{42}$$

Consider a node $j \in S_v(i)$ and intervene on

$$\mathcal{I} = \{j\} \cup \hat{\mathrm{pa}}(i) \cup \mathcal{B}(i) = \{j\} \cup \bigcup_{k=1}^{v-1} S_k(i) \cap \mathrm{pa_m}(i) \,. \tag{43}$$

We will show that

$$j \in \mathrm{pa_m}(i) \quad \Longleftrightarrow \quad X_j \not\!\perp\!\!\!\perp X_i \text{ in } p_{\mathrm{m},\mathcal{I}} \,. \tag{44}$$

If $j \in \mathrm{pa_m}(i)$, then by $\mathcal{I}$-mixture faithfulness, $X_j \not\!\perp\!\!\!\perp X_i$ in $p_{\mathrm{m},\mathcal{I}}$. We will prove the other direction, that is $X_j \not\!\perp\!\!\!\perp X_i$ in $p_{\mathrm{m},\mathcal{I}}$ implies $j \in \mathrm{pa_m}(i)$, similarly to the base case. First, note that $X_j \not\!\perp\!\!\!\perp X_i$ in $p_{\mathrm{m},\mathcal{I}}$ does not have a conditioning set. Then, it implies that there exists an active path $j \overset{\ell}{\rightsquigarrow} i$ in $\mathcal{G}_{\ell,\mathcal{I}}$ for some $\ell \in [K]$. Now, suppose that $j \notin \mathrm{pa_m}(i)$, which implies that the active path has the form $j \overset{\ell}{\rightsquigarrow} k \overset{\ell}{\rightarrow} i$ in $\mathcal{G}_{\ell,\mathcal{I}}$ for some $\ell \in [K]$ and $k \notin \mathcal{I}$. Since $k \in \mathrm{pa_m}(i)$, $k \notin \mathcal{I}$ implies that $k \notin \bigcup_{u=1}^{v-1} S_u(i)$. Then, we have $k \in \mathrm{de}_i(j) \cap \mathcal{A}$, which contradicts with $k \in S_v(i)$ due to definition of $S_v(i)$. Therefore, for $j \in S_v(i)$ and $\mathcal{I} = \mathcal{B}(i) \cup \hat{\mathrm{pa}}(i) \cup \{j\}$, $X_j \not\!\perp\!\!\!\perp X_i$ in $p_{\mathrm{m},\mathcal{I}}$ implies that $j \in \mathrm{pa_m}(i)$, which concludes the proof of the induction step. Therefore, by induction, the algorithm identifies $S_u \cap \mathrm{pa_m}(i)$ correctly for all $u \in \{1,\ldots,t\}$.

Finally, note that while processing each layer $S_u(i)$, the algorithm uses $|S_u(i)|$ interventions, one for each $j \in S_u(i)$, with size $|\mathcal{I}| = \left| \mathrm{pa_m}(i) \cap \bigcup_{k=1}^{u-1} S_k(i) \right| + \mathcal{B}(i) + 1$, where $\tau_i = |\mathcal{B}(i)|$ is referred to as the cyclic complexity of node $i$. Therefore, the size of the largest intervention set is

$$|\mathrm{pa_m}(i)| + \tau_i + 1 \,. \tag{45}$$

We note that this upper bound on the intervention size is $\tau_i$ larger than the necessary size $|\mathrm{pa_m}(i)| + 1$ shown in Theorem 1. This optimality gap reflects the effect of the cyclic complexity of the problem. Finally, adding $n$ single-node interventions performed in Step 1, for identifying $\mathrm{pa_m}(i)$ for all $i \in \mathbf{V}$, Algorithm 1 uses a total of

$$n + \sum_{i=1}^{n} |\mathrm{an_m}(i)| + \sum_{i=1}^{n} \sum_{u=1}^{t} |S_u(i)| = n + 2\sum_{i=1}^{n} |\mathrm{an_m}(i)| \leq 2n^2 - n = \mathcal{O}(n^2) \tag{46}$$

interventions, which completes the proof of the theorem.

# D  Additional examples

## D.1  Partitioning true edges into component DAGs

We have emphasized in Section 1 that the component DAGs of the mixture cannot be identified without further assumptions even under our interventional setting. We discuss a few examples of this matter.

1. Consider the following two mixtures of $K = 2$ DAGs

   - Mixture 1: Edge sets $\mathbf{E}_1 = \{1 \rightarrow 2, 1 \rightarrow 3\}$ and $\mathbf{E}_2 = \emptyset$
   - Mixture 2: Edge sets $\mathbf{E}_1 = \{1 \rightarrow 2\}$ and $\mathbf{E}_2 = \{1 \rightarrow 3\}$

   In this case, distributions of the two mixtures can still be the same under all intervention sets $\mathcal{I} \subseteq \{1, 2, 3\}$. Hence, without additional assumptions (e.g., model parameterization), we cannot distinguish the two mixtures via only conditional independence tests. Instead, we can only learn the set of true edges, $\mathbf{E}_t = \{1 \rightarrow 2, 1 \rightarrow 3\}$ for both mixtures.

2. Consider a mixture of two DAGs with edge sets $\mathbf{E}_1 = \{1 \rightarrow 2, 2 \rightarrow 3\}$ and $\mathbf{E}_2 = \{3 \rightarrow 1\}$. Recall that in Stage 1 of Algorithm 1, we learn the mixture ancestors of the nodes as an intermediate step. Hence, after learning the set of true edges in the mixture via the rest of the algorithm, we have the following information:

$$1 \in \mathrm{an_m}(3)\,, \quad 1 \notin \mathrm{pa_m}(3), \quad 1 \in \mathrm{pa_m}(2)\,, \tag{47}$$
$$2 \in \mathrm{pa_m}(3)\,, \quad 2 \notin \mathrm{an_m}(1)\,, \tag{48}$$
$$3 \in \mathrm{pa_m}(1)\,, \quad 3 \notin \mathrm{an_m}(2) \tag{49}$$

   Suppose that we know $K = 2$. Then, we can see that the only possible component DAGs that result in this mixture are $\mathbf{E}_1 = \{1 \rightarrow 2, 2 \rightarrow 3\}$ and $\mathbf{E}_2 = \{3 \rightarrow 1\}$. This is because $3 \rightarrow 1$ cannot be in the same DAG as the other two edges due to the known ancestral relationships. Hence, we learn the component DAGs in this case. However, without learning the true edges, we would not be able to know that we can learn the component DAGs of the mixture model.

These examples show that our work is a necessary first step into the interventional causal discovery of mixtures. Furthermore, we hope that it can inspire future work for the use of interventions in a mixture of models, e.g., establishing graphical conditions for (partial) recovery of individual DAGs, leveraging the side information.

Finally, we note two things regarding the possible side information that can enable stronger results. First, mixture distributions (the underlying component distributions $p_\ell(x)$s can be identified under some assumptions, e.g., Gaussian mixture models [47]. Second, in a related line of work, disentangling mixtures of unknown interventional datasets is studied under specific conditions on the intervention sets and given the distribution of the observational DAG [48]. Establishing the necessary and sufficient conditions for achieving similar disentangling objectives tasks in our mixture model is an open problem for future work.

## D.2  An example of cyclic complexity

We have empirically quantified the average cyclic complexity in Section 5. In addition, we give a visual example here. Consider the mixture of two DAGs in Figure 3. By definition of mixture ancestors, we have

$$\mathrm{an_m}(1) = \{2, 5\}\,, \tag{50}$$
$$\mathrm{an_m}(2) = \{3, 5\}\,, \tag{51}$$
$$\mathrm{an_m}(3) = \emptyset\,, \tag{52}$$
$$\mathrm{an_m}(4) = \{1, 2, 3, 5\}, \tag{53}$$
$$\mathrm{an_m}(5) = \{1, 2\}\,. \tag{54}$$

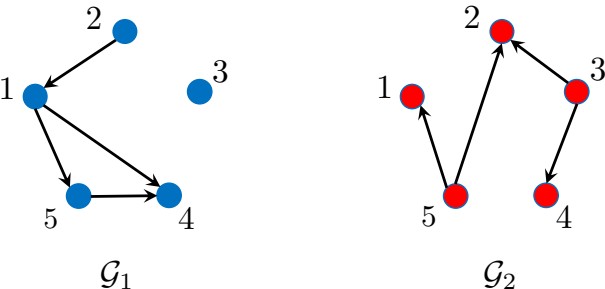

Figure 3: Sample DAGs for a mixture of two DAGs

Then, by definition of $\mathcal{C}(i)$ in (10), we have

$$\mathcal{C}(1) = \{(2,5,2)\}, \tag{55}$$
$$\mathcal{C}(2) = \emptyset, \tag{56}$$
$$\mathcal{C}(3) = \emptyset, \tag{57}$$
$$\mathcal{C}(4) = \{(2,5,2),(2,1,5,2),(1,5,1)\}, \tag{58}$$
$$\mathcal{C}(5) = \emptyset. \tag{59}$$

Subsequently, an example of minimal breaking sets and cycle complexities are given by

$$\mathcal{B}(1) = \{2\} \implies \tau_1 = 1, \tag{60}$$
$$\mathcal{B}(2) = \emptyset \implies \tau_2 = 0, \tag{61}$$
$$\mathcal{B}(3) = \emptyset \implies \tau_3 = 0, \tag{62}$$
$$\mathcal{B}(4) = \{5\} \implies \tau_4 = 1, \tag{63}$$
$$\mathcal{B}(5) = \emptyset \implies \tau_5 = 0. \tag{64}$$

This example illustrates that even though the mixture model can contain many cycles, the cyclic complexity of the nodes can be small.

# E   Additional experiments

**Effect of the number of samples.**   For the same experimental setting in Section 5, we investigate the effect of the number of samples on the performance of Algorithm 1. Figure 4a demonstrates that the algorithm achieves almost perfect precision even with as few as $s = 1000$ samples under the parameterization described in Section 5. The recall rates are lower than the precision; however, when the number of samples is increased to $s = 10000$, the gap is closed, and the recall rates also become closer to perfect.

**Varying the true edge weights.**   Recall that in Section 5, we have considered the case where the weight of a true edge is constant across the component DAGs it belongs to. However, our theory and algorithm can handle the general case, in which conditional distributions $p_\ell(X_i \mid X_{\mathrm{pa}_\ell(i)})$ and $p_{\ell'}(X_i \mid X_{\mathrm{pa}_{\ell'}(i)})$ can be different even if the parent sets $\mathrm{pa}_\ell(i) = \mathrm{pa}_{\ell'}(i)$ for two component DAGs $\mathcal{G}_\ell$ and $\mathcal{G}_{\ell'}$. For instance, when considering a mixture of DAGs where $(1 \to 2) \in \mathbf{E}_1$, $(1 \to 2) \notin \mathbf{E}_1$, and $(1 \to 2) \in \mathbf{E}_3$ relations, the edge weight from node 1 to node 2 can be different in $\mathcal{G}_1$ and $\mathcal{G}_3$. To investigate the performance of our algorithm in this general setting, we consider the following parameterization. When randomly generating the weight of a true edge $(i \to j) \in \mathbf{E}_t$, it has two options: (i) With probability $0.5$, it is constant across the component DAGs it belongs to, (ii) with probability $0.5$, it is different (randomly sampled) for every component DAG it belongs to. Figure 4b shows that the performance of the algorithm is virtually the same for this setting compared to the main setting considered in Section 5.

**Skeleton (true edges) versus inseparable pairs.**   In Section 5, we demonstrate the need for interventions by learning inseparable pairs from observational mixture data and comparing them to the skeleton of the true edges. To learn the inseparable pairs specified in (5), we perform exhaustive conditional independent tests as in [11], summarized in Algorithm 2. Note that as mentioned in Section 4.1, this exhaustive search requires $\mathcal{O}(n^2 \cdot 2^n)$ CI tests, which we perform only for this experiment setting and omit in our proposed Algorithm 1.

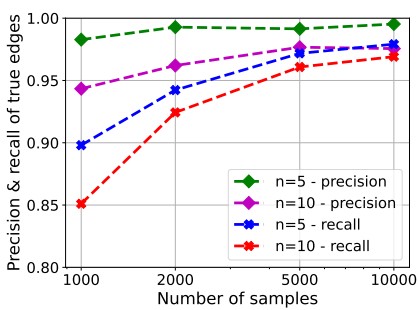
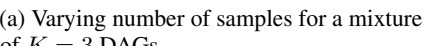
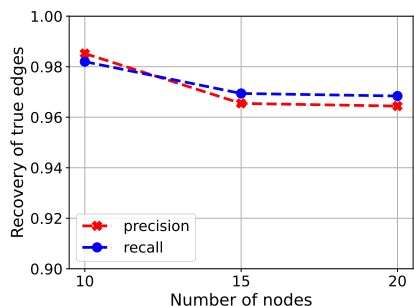

(a) Varying number of samples for a mixture of $K = 3$ DAGs

(b) Recovery rates for varying true edge weights for a mixture of $K = 3$ DAGs

Figure 4: Additional experiment results for true edge recovery

---

**Algorithm 2** Mixture skeleton learning via observational data [11, Algorithm 1 - Stage 1]

---

1: **Input:** Samples from mixture distribution $p_{\mathrm{m}}$

---

2: **Return: Inseparable pairs of the mixture** $\mathbf{E}_{\mathrm{i}}$
3: Form complete undirected graph: $\mathbf{E}_{\mathrm{i}} \leftarrow \{(i - j) : i, j \in \mathbf{V}\}$
4: **for** all $i, j \in \mathbf{V}$ **do**
5:     **for** all $S \in \mathbf{V} \setminus \{i, j\}$ **do**
6:         **if** $X_i \perp\!\!\!\perp X_j \mid X_S$ **then**
7:             Remove $(i - j)$ edge: $\mathbf{E}_{\mathrm{M}} \leftarrow \mathbf{E}_{\mathrm{M}} \setminus (i - j)$, and $\mathrm{SepSet}(i, j) \leftarrow S$.
8:             **break**
9:         **end if**
10:     **end for**
11: **end for**

---

