# OpenReview forum: "Interventional Causal Discovery in a Mixture of DAGs"
_NeurIPS.cc/2024/Conference — NeurIPS 2024 poster_

### Official Review · Reviewer_Tiwj · 2024-07-11

**Soundness:** 3
**Presentation:** 3
**Contribution:** 3
**Rating:** 5
**Confidence:** 2

**Summary:**

This paper deals with using interventions to learn the causal links in a mixture-of-DAGs model.
They find the minimum number of intentions that are required to learn "true edges" where a true edge from a node X to a node Y indicates that at least in one mixture component, X is a parent of Y. They also present an algorithm that learns true edges via interventions. They show that in terms of the number of interventions, the proposed algorithm is close to optimal.

**Strengths:**

1. Up to my knowledge this is the first time that the problem of using interventions to learn the mixture of causal DAGs is studied.
2. As far as I could follow, the presented theorems on the number of sufficient and necessary interventions as well as the proposed algorithm are correct.

**Weaknesses:**

1. To present their theorems and algorithm, they require several definitions and this can be confusing in the first read. If they could provide a simple running example (i.e. a mixture of DAGs) by which they could illustrate/clarify all definitions, the paper would be easier to follow --in my opinion.

2. The provided experimental results are on synthetic data only. This, I think is the main weakness of this paper. If (as claimed in the abstract and intro) many causal systems are mixtures of DAGs, it would be helpful if the authors could present and analyze at least one such real-world system in the experiments section.

**Questions:**

Is there any existing and open-source real-world dataset where the data generation mechanism is a mixture of DAGs? If yes, is there a reason that such a dataset is not utilized in this paper?

---

> ### Author Rebuttal · Authors · 2024-08-06
>
> We thank the reviewer for thoughtful evaluations and feedback. We hope the following explanations can clarify the reviewer’s concerns.
>
> **Running example:** Thank you for the suggestion. Figure 2 in Appendix D illustrates the mixture of DAGs and construction of $\cal{I}$-DAG. Due to space limitations, we have had to defer it to the appendix. We will move it to the main paper and add the following details to its caption:
> - True edges $E_{\rm t} = \\{1 \rightarrow 2, 2 \rightarrow 3, 3 \rightarrow 2, 3 \rightarrow 4, 1 \rightarrow 4\\}$
> - Emergent pairs $E_{\rm e} = \\{ (1,3), (2,4) \\}$
> - Inseparable pairs $E_{\rm i} = \\{ (1,2), (1,3), (1,4), (2,3), (2,4), (3,4) \\}$
> - $\Delta$-through path examples: $1 \rightarrow 2 \rightarrow 3$ in $\mathcal{G}_1$; $3 \rightarrow 4$ in $\mathcal{G}_2$
>
>
> **Real-world dataset:** We emphasize that the focus of our work is establishing the needed *theory* for using interventions on a mixture of DAGs. While we acknowledge the importance of real-world applications, establishing the novel theory *and* demonstrating the real-data usefulness can be beyond the scope of one paper. In the following, we elaborate on the challenges of a direct real-world application and explain why our theoretical contribution is an important step toward that direction.
>
>
> - **Lack of a proper real-world dataset:**   The existing benchmark causal discovery datasets, such as protein signaling network in Sachs et al. (2005) or ovarian cancer dataset of Tothill et al. (2008), are known to be *not* perfect single DAGs. For instance, the “true DAG” for protein signaling network is challenged under new evidence (Ness et al., 2017), and there is no strong consensus over the ground truth DAG. Furthermore, the existing interventional single DAG learning algorithms do not perform well in terms of standard metrics, e.g., structural Hamming distance (e.g., Squires et al. (2020), Wang et al. (2018), Varici et al. (2021)) despite the strong theoretical guarantees. These observations suggest that a single DAG may not be the best approach for modeling such real-world datasets. To our knowledge, there is no commonly accepted and well-defined dataset in which the data generation mechanism is a mixture of DAGs.
>
> - **Access to interventions:**  We note that the existing datasets come with a predefined set of interventions, e.g., single-node or two-node genomics interventions, and we cannot "simulate" any desired intervention. This makes it difficult to evaluate any interventional learning algorithm that fully learns the graph. For instance, as discussed in Lines 84—87, the majority of the existing results in the single-DAG setting consider “unconstrained” intervention sizes. Our results in Theorem 1 and 2 contribute to our understanding of the *required* intervention sizes, which can shed light on the extent of identifiability given a real-world dataset (without a simulator).
>
> In light of these, we leave the investigation of real-world datasets/applications (and related partial identifiability results) to future work.
>
> **References**
>
> K. Sachs, O. Perez, D. Pe'er, D. A. Lauffenburger, and G. P. Nolan, “Causal protein-signaling networks derived from multiparameter single-cell data,” Science, vol. 308, no. 5721, pp. 523– 529, 2005.
>
> R. W. Tothill et al. “Novel molecular subtypes of serous and endometrioid ovarian cancer linked to clinical outcome,” Clinical Cancer Research, vol. 14, no. 16, pp. 5198–5208, 2008.
>
> R. O. Ness, K. Sachs, P. Mallick, and O. Vitek, “A Bayesian active learning experimental design for inferring signaling networks,” in Proc. Research in Computational Molecular Biology, Hong Kong, May 2017, pp. 134–156.
>
> C. Squires, Y. Wang, and C. Uhler, “Permutation-based causal structure learning with unknown intervention targets,” in Proc. Conference on Uncertainty in Artificial Intelligence, August 2020.
>
> Y. Wang, C. Squires, A. Belyaeva, and C. Uhler, “Direct estimation of differences in causal graphs,” in Proc. Advances in Neural Information Processing Systems, Montreal, Canada, December 2018.
>
> B. Varici, K. Shanmugam, P. Sattigeri, and A. Tajer. “Scalable intervention target estimation in linear models,” in Proc. Advances in Neural Information Processing Systems, December 2021.

---

> > ### Comment · Reviewer_Tiwj · 2024-08-12
> >
> > Thanks for the response

---

### Official Review · Reviewer_2inT · 2024-07-12

**Soundness:** 3
**Presentation:** 3
**Contribution:** 3
**Rating:** 6
**Confidence:** 4

**Summary:**

The paper studies the setting where data is generated from a mixture of DAGs and one wishes to recover the "true edges" (edges that exist in at least one of the underlying DAGs). Similar to the usual causal discovery setting, observational data alone is insufficient and interventions are required. The paper characterize the set of interventions needed to learn these true edges and provide an algorithm for doing so. Some small scale experiments and source code were given.

**Strengths:**

The concept of mixture of DAGs provide an alternative framework to capture cyclic causal system and, to my understanding, is under-explored. This paper aims to fill some of these gaps.

**Weaknesses:**

The paper strives to recover "true edges" (objective 1 on Line 195) but I don't think they can map these edges to the appropriate underlying DAG within within the mixture of DAGs. It is unclear why it is interesting to be able to know these edges without the context of the DAG they belong to. To me, this severely weakens the motivation/usefulness of this work.

The experimental evaluation felt rather weak.

**Questions:**

General questions:
- [1] also studied the universal lower bound problem that you mentioned on Line 92, and [2] eventually provided an exact characterization for the number of interventions required to recover the DAG from the observational essential graph. You may want to consider adding these references in your revision.
- Can you motivate why it is sufficient/useful enough to recover "true edges" (objective 1 on Line 195)? Your approach cannot identify which DAG each of the recovered edges belong to within the mixture of DAGs, right?
- Is the number $K$ of mixture models given to the algorithm as input? The phrase "a priori knowledge of the number of mixture components" on Line 251-252 seem to suggest that $K$ is typically not given. Also, Algorithm 1 is independent of $K$. However, $K$ features prominently in the characterization (e.g. Theorem 2), so how does one perform the correct set of interventions without $K$ as input?
- On Line 282, you mention "intervening on any set $I$ that contains $B(i)$ breaks all the cyclic relationships in $C(i)$". Why is this so? I thought some "edge cutting" (i.e. exactly one endpoint of the edge is in intervention set) is needed in order to distinguish edge directions? For example, intervening on the entire set of vertices will trivially include $B(i)$ but this does not provide any useful information, right?
- In Line 290, do you want to perhaps write $(A \setminus (S_1(i) \cup \ldots \cup S_{n-1}(i)))$ since the algorithm is "removing the layer $S_u(i)$ from $A$ after each iteration", or at least reference Line 26 in Algorithm 1?
- In equations (23) and (24) of Appendix C, shouldn't are you missing $\cup \{i\}$? I don't think you defined descendants and ancestors to include $i$ itself, or maybe I'm mistaken?
- On Line 620, you wrote $S_1$. I don't see this defined anywhere (please correct me if I missed it). I don't think this is a typo of $S_1(i)$ here since $j \in S_1(i)$ and $j \not\in pa_m(j)$. This undefined notation also appears in the remaining of this proof and is inhibiting my ability to verify Theorem 3.
- The mixture in Figure 2 assumes that $P_{G_1}(1) = P_{G_2}(1)$ since the $y$ node does not point to the $I$-mixture DAG (definition 5 and equation (2)), right? This assumption was not mentioned in Appendix D.

Experiment questions:
- The graph sizes in the experimental evaluation were extremely small. Is there a reason for this? In Appendix E (line 727), you say "only a marginal decrease in performance" when $K = \{2,3,4\}$ and $n \in \{5,\ldots,10\}$. I'm not sure if that is enough scale to draw such a conclusion...
- By setting p = 2/n in the Erdos-Renyi G(n,p) graph generation, the graphs will likely be disconnected (random graph theory tells us we need about p ~= log n / n for the graph to be connected). Will this be a problem for your experiments if you scale to larger graphs?
- Consider mentioning the edge weights of your linear Gaussian model in the main paper instead of the appendix. Line 348 claims that s = 1000 samples are sufficient for almost perfect precision but finite sample guarantees for "good recovery of the graph" scale non-trivially with the correlation strength, e.g. see the finite sample analysis for linear Gaussians in [3].
- What is the mixture skeleton learning algorithm?
- In Appendix E, you mentioned that "the same edge weight is assigned to all realizations of a true edge across all component DAGs". Why is this reasonable? I thought the whole point of mixture models is that $p_\ell \neq p_{\ell'}$ in general? See Line 117.

Possible typos:
- Extra bold of "ixture" on Line 255?
- Equation (57): Do you mean $C(5) = \emptyset$?

References:

[1] Porwal, Vibhor, Piyush Srivastava, and Gaurav Sinha. "Almost Optimal Universal Lower Bound for Learning Causal DAGs with Atomic Interventions." International Conference on Artificial Intelligence and Statistics. PMLR, 2022.

[2] Choo, Davin, Kirankumar Shiragur, and Arnab Bhattacharyya. "Verification and search algorithms for causal DAGs." Advances in Neural Information Processing Systems 35 (2022): 12787-12799.

[3] Kalisch, Markus, and Peter Bühlman. "Estimating high-dimensional directed acyclic graphs with the PC-algorithm." Journal of Machine Learning Research 8.3 (2007).

**Limitations:**

Nil

---

> ### Author Rebuttal · Authors · 2024-08-06
>
> We are grateful for the exceptionally detailed and thoughtful review. We address the raised questions as follows.
>
> ## General questions
>
> **Motivation for recovering the true edges**: The observation is correct that, without further assumptions, we cannot identify which true edges belong to which DAGs. To see why this is impossible in general, consider two mixtures:
> -    Mixture 1: $K=2$ DAGs with edges $E_1=\\{1\rightarrow 2, 1\rightarrow 3\\}$ and $E_2=\emptyset$
> -    Mixture 2: $K=2$ DAGs with edges $E_1=\\{1\rightarrow 2\\}$ and $E_2=\\{1\rightarrow 3\\}$
> In this example, no intervention can distinguish the two mixtures.
>
> Then, if we cannot disentangle the individual DAGs, we argue that finding the true edges is the best we can do, and it is still useful for causal inference tasks. The motivation is that the mixture model is generally composed of DAGs with similar contexts. For instance, different subtypes of ovarian cancer create a mixture model (Lines 27-28). Identifying a causal connection (a true edge) in this setting is crucial even if it appears in only some subpopulations (i.e., some of the component DAGs).
>
> **Number of mixture components $K$ is unknown**: The algorithm does not take $K$ as an input. Our algorithm is designed for general DAGs (without structural restrictions). Hence, the guarantees associated with the algorithm are given for general DAGs in Theorems 3 and 4. The mentioned Theorem 2 is specific to trees and aims to provide a theoretical understanding separate from the algorithmic approach.
>
> **Breaking set in Line 282**:
> -    First, we elaborate on the procedure. When considering an intervention on $B(i)$, we already have the ancestor set $\hat{\rm an}(i)$. Hence, for $j\in\hat{\rm an}(i)$, we investigate the *single* direction $j\rightarrow i$. Since $i\notin B(i)$, we avoid making a useless intervention, e.g., intervening on both $i$ and $j$ at the same time.
> -    Next, recall the definition of $C(i)$: set of cycles in which all nodes of the cycle belong to $\hat{\rm an}(i)$. By definition, $B(i)$ contains at least one node from each cycle in $C(i)$. Therefore, by intervening on $B(i)$, we cut off a link in each cycle.
>
> **Additional references**: Thanks! Indeed, both [1] and [2] are related to the discussion in our literature review. We will add them to the revised paper.
>
> **Example in Figure 2**: Your observation is correct. For simplicity, we have used an example in which the root node $1$ has the same marginal distribution on two components. We will mention it in the caption of Figure 2.
>
> **Typos and minor fixes**:
>
> -   **Proof of Theorem 3**: We are truly sorry for the typos here. In Line 620, it should be $S_1(i)$ *and* ${\rm pa}_{\rm m}(i)$. Since Theorem 3 considers a single node $i$, all $S_t$ sets are supposed to be $S_t(i)$. The entire list of fixes:
>     -   $S_1\to S_1(i)$ in Lines 620, 628, 629, and eq. (26)
>     -   $S_u\to S_u(i)$ in Lines 633, 635, 643
>     -   $S_k\to S_k(i)$ in Line 634.
>
> -   **Line 290**: Thanks for the suggestion. We update Lines 289-293 as follows:
>
>     *Next, we update $\mathcal{A}\gets\mathcal{A}\setminus S_1(i)$ by removing layer $S_1(i)$ to conclude the first step. Then, we iteratively construct the layers $S_u(i)=\\{j \in \mathcal{A}:\hat{\rm de}(j)\cap\mathcal{A}=\emptyset\\}$ and update $\mathcal{A}\gets\mathcal{A}\setminus S_u(i)$ as in Line 26 of the algorithm. We continue until the set $\cal{A}$ is exhausted, and denote these topological layers by $\\{S_1(i),\dots,S_t(i)\\}$.*
> -   **Eq.(23) and (24)**: You are right that descendant and ancestor definitions *do not* include $i$. Hence, Eq.(23) and (24) should have $\cup i$, as correctly given in line 14 of the algorithm. Correcting another typo in Eq.(24), we have
>     ${\rm de}_i(j)=\hat{\rm de}(j)\cap\\{\hat{\rm an}(i)\cup i\\}={\rm de}\_{m}(j) \  , \ \forall j \in{\rm an}\_{\rm m}(i)\ .$
> -   **Eq.(57)**: You are right that we mean $\mathcal{C}(5) = \emptyset$, we will fix it.
> -   **Line 255**: It should be **M**ixture.
>
> ## Experiment questions:
>
> **Increasing graph size and number of components**: Please refer to the global response in which we report experiments up to $n=30$ nodes and $K=10$ components.
>
> **Connectedness of the graph**: Our theory and algorithm do not require the DAGs to be connected. So it wouldn't be a problem for experiments. As the reviewer points out, large random graphs with small densities would consist of a giant connected component and many small components. In this case, we’d expect the "giant connected component" to dominate the complexity of our algorithm (e.g., the number of total interventions).
>
> **Edge weights**: We will move the “experimental procedure” paragraph in Appendix E to the main paper and will add the statement “under this parametrization” to the discussion on the number of samples in Line 348.
>
> **Parametrization of true edges**: You are right that in general, conditional distributions $p_l(X_i\mid X_{{{\rm pa}\_{l}}(i)})$ and $p_{l'}(X_i\mid X_{{{\rm pa}\_{l'}}(i)})$ can be different even if ${\rm pa}_l(i)={\rm pa}\_{l'}(i)$. For simplicity of the exposition of experiments, we considered the special case of mixtures in which a change in the conditional distribution is only caused by changes in the parents so that $\Delta$ becomes the set of nodes with varying parents across the component DAGs. We perform additional experiments for the most general case where true edges can have different weights across the components. Please see the global response for the results and details.
>
> **Mixture skeleton learning**: It is done via exhaustive CI tests. For every pair $(i,j)$, we test the conditional independence of $X_i$ and $X_j$ given every $S\subseteq[n]\setminus\\{i,j\\}$ (as in Algorithm 1 of [11]). We will add this detail to the experiments in the revised paper. Note that, as mentioned in Line 262, we omit this step due to its $\mathcal{O}(n^2 2^n)$ complexity and only perform it for the comparison in Figure 1b.

---

> > ### Comment · Reviewer_2inT · 2024-08-09
> >
> > Thank you for your detailed responses! I am very satisfied with them and intend to maintain my positive score.
> >
> > I have a non-technical follow-up question that I am curious to hear from the authors.
> >
> > **Recovering true edges**:
> >
> > I understand and can appreciate that in certain mixtures, just learning the direction of 1 true edge is sufficient to have a real-world impact. However, in the example of your response, the edges are (i) "consistent" and (ii) acyclic. What if there is a mixture such that
> >
> > (i) $1 \to 2$ in one DAG and $2 \to 1$ in another, and
> >
> > (ii) $1 \to 2$ and $2 \to 3$ in one DAG while $3 \to 1$ in another
> >
> > In such scenarios, how should one make use of the recovered edges in a meaningful manner?

---

> > > ### Author Response · Authors · 2024-08-09
> > >
> > > We are glad to hear that our response addressed your questions! We comment on your follow-up examples as follows.
> > >
> > > - Having cyclic relationships in your examples fits the motivation of mixture models well. Consider your first example with $\mathcal{G}_1 : 1 \rightarrow 2$ and $\mathcal{G}_2 : 2 \rightarrow 1$. In this case, learning the true edges tells us about what we *should not do*, more than what to do. Suppose that an experimenter does not know which model they should adopt for the data; a DAG or a mixture of DAGs. If they elect a single DAG with a single edge, say $1 \rightarrow 2$ (possibly, the stronger among the two ground truth edges), there may be unintended consequences. For instance, if the mixture model is due to a feedback loop, then trying to control the level of $X_2$ via $X_1$ will result in an unintended increase in $X_1$. On the other hand, by taking the cautious route and considering a mixture mode, we identify the true edges in both directions and avoid making incorrect inferences.
> > >
> > > - The second example is a very interesting case. Note that, in Stage 1 of the algorithm, we learn the "mixture ancestors" as an intermediate step. Hence, we have the information $1 \in {\rm an}\_{\rm m}(3), 1 \notin {\rm pa}\_{\rm m}(3), 2 \in {\rm pa}\_{\rm m}(3), 3 \in {\rm pa}\_{\rm m}(1), 2 \notin {\rm an}\_{\rm m}(1), 3 \notin {\rm pa}\_{\rm m}(2), 1 \in {\rm pa}\_{\rm m}(2)$. Suppose that we know $K=2$. Then, we can see that the only possible mixture is $E_1 = \\{1 \rightarrow 2, \ 2 \rightarrow 3\\}$ and $E_2 = \\{3 \rightarrow 1\\}$, so we learn the individual DAGs in this case! (this is because $3 \rightarrow 1$ cannot be in the same DAG as either of the other two edges due to the known ancestral relationships). Without learning the true edges though, we cannot say whether we can learn the individual DAGs from the mixture model.
> > > - The second example also shows that our work can inspire future work for the use of interventions in a mixture of models, e.g., establishing graphical conditions for (partial) recovery of individual DAGs, leveraging the knowledge of $K$ when provided.

---

> > > > ### Comment · Reviewer_2inT · 2024-08-10
> > > >
> > > > I see! Thanks for the responses.

---

### Official Review · Reviewer_bGit · 2024-07-13

**Soundness:** 3
**Presentation:** 4
**Contribution:** 4
**Rating:** 7
**Confidence:** 3

**Summary:**

This work studies an important problem in causal discovery for its relevance in the real world -- identifying the causal relationship when the underlying data-generating process comes from a mixture of different DAGs. They give the necessary and sufficient size of intervention set to identify the union of all the parents of a node across components. They also give an algorithm that requires $O(n^{2})$ interventions to identify all the directed edges of individual DAG. Finally, they also quantify the gap between the number of interventions used by the proposed algorithm and the optimal size in terms of cyclic complexity number.

**Strengths:**

1. The paper is well-written and easy to follow.
2. Studying a mixture of DAG is underexplored but will help bring the application of causality closer to the real world. This paper furthers the line of work in this direction and is thus important.
3. This paper characterizes the necessary and sufficient conditions to identify the "true" edges, i.e., the edges that are actually present in at least one of the components of the mixture from the emergent edges. Also, they don't impose restricting assumptions on component DAGs like poset compatibility from previous work, thereby generalizing the results to a richer mixture family.
4. Also, to the best of my knowledge, this is the first work in causal discovery for a mixture of DAGs that allows for the use of interventional data, thereby improving identifiability.

**Weaknesses:**

1. It is understandable that this paper doesn't show the sample complexity of identifying the true edge set, but maybe the empirical section could be more diverse with a larger number of nodes and components to get a sense of the proposed algorithm's statistical efficiency.

**Questions:**

1. This is not directly relevant to the proposed algorithm and might be a direction for future work.  But is it possible to partition the identified true edges into subsets for each of the individual components? Can we even identify the number of components in the mixture?

**Limitations:**

Yes

---

> ### Author Rebuttal · Authors · 2024-08-06
>
> We thank the reviewer for the thoughtful comments and assessment of our paper. We address the raised questions as follows.
>
> **Experiments**: In the additional experiments in the global response, we demonstrate that our algorithm is scalable to higher dimensions – up to $n=30$ nodes and $K=10$ component DAGs – without a significant change in the performance. Please refer to the global response for details.
>
>
> **Partitioning true edges into individual component DAGs**: Thank you for the suggestion that is an important future direction. However, it will most certainly require additional assumptions. For instance, in our current formulation, consider two sets of mixture DAGs:
> -    *Setup 1*: $K=2$ DAGs with edge sets $E_1 = \\{1\rightarrow 2, 1\rightarrow 3\\}$ and $E_2 = \emptyset$
> -    *Setup 2*: $K=2$ DAGs with edge sets $E_1 = \\{1\rightarrow 2\\}$ and $E_2 = \\{1\rightarrow 3\\}$
> In this example, no interventions can distinguish the two setups. Similarly, in general, we cannot determine the number of components in the mixture.
>
> That being said, under certain assumptions we expect the partitioning to become possible. For instance, in a similar problem, Kumar and Sinha (2021) study disentangling mixtures of unknown interventional datasets under specific conditions on the intervention sets and given the distribution of the pre-intervention DAG. Establishing the necessary and sufficient conditions for achieving similar disentangling objectives tasks in our mixture model is an open problem for future work.
>
> - Kumar, Abhinav, and Gaurav Sinha. "Disentangling mixtures of unknown causal interventions." Uncertainty in Artificial Intelligence. PMLR, 2021.

---

> > ### Comment · Reviewer_bGit · 2024-08-12
> >
> > I thank the author for their response. I will maintain my score.

---

### Official Review · Reviewer_5C9J · 2024-07-15

**Soundness:** 4
**Presentation:** 3
**Contribution:** 3
**Rating:** 7
**Confidence:** 3

**Summary:**

In the case of a single DAG, conditional independence tests specify the skeleton (under faithfulness) and interventions are limited to orienting edges. In the case of data coming from a mixture of DAGs, it is possible for two variables to not be adjacent in any of the components but still be conditionally dependent for every conditioning set. This paper proves that the necessary and sufficient intervention size to identify the "true" edges of the mixture, i.e. node pairs where there is an edge in at least one component DAG is the size of the union of parents in all components + 1. It also designs an algorithm to do so.

**Strengths:**

Causal discovery given data from a mixture of DAGs seems to be an important problem with applications in several domains. Existing work has focused on the observational data regime. But this is not sufficient in the mixture case because of the phenomenon of "emergent pairs". This is the first paper that studies this problem assuming presence of interventional data. Using interventional data to pinpoint DAGs  among an equivalence class is also a relevant thread of interest.
The paper is written well with a clear flow of ideas. I didn't check all the proofs but the ones that I didn't, appear sound. The analysis of the problem appears quite complete in a sense with the optimal intervention sizes characterized and the algorithm's gap to optimality also characterized.

**Weaknesses:**

The results heavily depend on the interventional model where the modified distribution of the intervened variable does not change across component DAGs. I was wondering if there was any practical motivation for considering such an intervention model. If so, it would make sense to include it in the paper.

I also could not find connections to existing results on intervention sizes for single DAG which should be a special case of thie mixture case. Is this because of the interventional model, again?

**Questions:**

Already asked in the weakness section.

**Limitations:**

Addressed implicitly in a conclusions section

---

> ### Author Rebuttal · Authors · 2024-08-06
>
> We thank the reviewer for a thorough review and insightful comments. We address the questions as follows.
>
> **Modified distributions of an intervened variable:** We considered an intervention model in which an intervened node $i$ has distribution $q_i(X_i)$ for all component DAGs. The reason is that an intervention procedure targets a specific node on all component models at the same time. For instance, when considering a gene knockout experiment (e.g., via CRISPR technique (Ran et al., 2013)), all the edges from parents are cut off, and the new distribution $q_i(X_i)$ is expected to be the same across all DAGs as result of the same intervention mechanism.
>
> F Ann Ran, Patrick D Hsu, Jason Wright, Vineeta Agarwala, David A Scott, and Feng Zhang. “Genome engineering using the crispr-cas9 system”. Nature Protocols, 8(11):2281–2308, 2013.
>
> **Intervention sizes for single DAGs:**
> - First we note that an assumption in our analysis is that we have $K \geq 2$ component DAGs. Hence, unfortunately we cannot recover the counterpart results for single DAGs as a special case. Furthermore, when considering a single DAG, there are no multiple component DAGs. As such, the lack of a detailed discussion on single DAGs is not due to our interventional model but it is simply due to the nature of the mixture model that does not subsume the single DAG model.
>
> - After these clarifications,  we kindly note that we discussed the interventions on single DAGs in Lines 83-98. To elaborate further, single DAGs in causally sufficient systems, i.e., no unobserved confounders, can be learned using **single-node** interventions. Hence, almost none of the papers cited in the paragraph "Intervention design for causal discovery of a single DAG" investigates the required intervention sizes. Therefore, we did not discuss the specific results of those papers that are not related to our investigation. To give an example [16] shows that $\cal{O}(\frac{n}{k}\log \log k)$ randomized interventions with size $k$ suffice for identifying the DAG with high probability. In another direction, different cost models are proposed to minimize the total intervention cost incurred by the number and size of the interventions ([20], [24], [25]). In the presence of latent variables, multi-node interventions can be required. However, for this case, [19]-[20]-[21] study strongly-separating sets, again based on unconstrained intervention sizes.
>
> - To our knowledge, only related work for the required size of interventions is [26]. Specifically, it considers **cyclic** directed models and shows that the required intervention size is at least $\zeta-1$ where $\zeta$ denotes the size of the largest strongly connected component (nodes $i$ and $j$ are said to be strongly connected if they are both ancestors of each other). We will add this note to the revised paper.

---

> > ### Comment · Reviewer_5C9J · 2024-08-13
> > **Response to rebuttal**
> >
> > Thanks for the response. I am happy to maintain my original score.

---

### Author Rebuttal · Authors · 2024-08-06

We thank all reviewers for their thorough evaluation and thoughtful questions. To demonstrate the scalability of our algorithm, we performed additional experiments under the same settings described in the paper.

**Increasing the number of nodes**: The submitted paper presents experiment results for up to $n=10$ nodes. In Figure 1a of the pdf attached to the global response, we report the average precision and recall rates (of 50 runs) for varying the number of nodes up to $n=30$ under a mixture of $K=3$ DAGs.

**Increasing the number of components in the mixture**: The submitted paper presents experiment results for up to $K=4$ component DAGs. In Figure 1b of the pdf attached to the global response, we report the average precision and recall rates (of 50 runs) up to $K=10$ component DAGs with $n=10$ nodes.

Both results show that there is only a marginal decrease in the performance of recovering the true edges which shows that our algorithm is scalable.

**Parameterization of true edges**:
We also performed additional experiments to demonstrate that the algorithm can handle true edges with different strengths across component DAGs. Specifically, in general, conditional distributions $p_l(X_i\mid X_{{{\rm pa}\_{l}}(i)})$ and $p_{l'}(X_i\mid X_{{{\rm pa}\_{l'}}(i)})$ can be different even if the parent sets are the same, ${\rm pa}_l(i) = {\rm pa}\_{l'}(i)$ (as mentioned in Line 117). For the experiments for this setting, we consider $n\in\\{10,15,20\\}$ nodes and $K=3$ DAGs. The weight of each true edge has two options:
-    With probability 0.5, it is fixed across the component DAGs it belongs to
-    With probability 0.5, it is different for every component DAG it belongs to.

Figure 2 in the attached pdf shows that the performance of the algorithm is virtually the same for this setting and the main setting we considered with fixed true edge weights.

---

### Decision · Program_Chairs · 2024-09-25

**Decision:**

Accept (poster)

**Comment:**

The paper is well received across the board overall, with most reviewers enthusiastic about the acceptance of the paper.

I-mixture faithfulness is swept under the rug a bit too quickly. Authors simply say they have to use this for learning from interventions. Please compare this proposed assumption with the more standard interventional faithfulness assumptions in the literature. Does one imply the other and vice verse? Please add a remark on this.

I appreciate the matching lower bounds authors provide.

Please also add the more explicit quantitative comparison and discussions related to single DAG learning results in the literature, as in your answer to Reviewer 5C9J.

Based on Reviewer bGit's and Reviewer 2inT's questions, please emphasize that the proposed method cannot identify individual DAGs without further assumptions and include the examples you provided in the rebuttal in the main paper. The goal of the paper and what "true edges" mean should be very clear from the start, which I think is not as explicit as it could be at times.